# WikiBigEdit: Understanding the Limits of Lifelong Knowledge Editing in LLMs

**Lukas Thede** [1 2 3]  **Karsten Roth** [1 2 3]  **Matthias Bethge** [1]  **Zeynep Akata** [2 3 4]  **Thomas Hartvigsen** [5]

## Abstract

Keeping large language models factually up-to-date is crucial for deployment, yet costly retraining remains a challenge. Knowledge editing offers a promising alternative, but methods are only tested on small-scale or synthetic edit benchmarks. In this work, we aim to bridge research into lifelong knowledge editing to real-world edits at a practically relevant scale. We first introduce `WikiBigEdit`; a large-scale benchmark of real-world Wikidata edits, built to automatically extend lifelong for future-proof benchmarking. In its first instance, it includes over 500K question-answer pairs for knowledge editing alongside a comprehensive evaluation pipeline. Finally, we use `WikiBigEdit` to study existing knowledge editing techniques' ability to incorporate large volumes of real-world facts and contrast their capabilities to generic modification techniques such as retrieval augmentation and continual finetuning to acquire a complete picture of the practical extent of current lifelong knowledge editing.[1]

## 1. Introduction

Keeping large language models factually up-to-date is a critical challenge in their practical deployment (Hartvigsen et al., 2023; Meng et al., 2023; 2022; Wang et al., 2024a; Zhang et al., 2024) due to given knowledge cutoffs and limited tail-end concept coverage (Cheng et al., 2024; Zheng et al., 2024; Zhang et al., 2024; Udandarao et al., 2024) incurring inaccuracies. Finding ways to inject or update new and revised facts is crucial; Particularly for factuality-critical applications like medicine (Bao et al., 2023; Jeong et al., 2024), law (Colombo et al., 2024; Lai et al., 2024) or education (Team et al., 2024c; Wang et al., 2024b).

However, frequent full model retraining is prohibitively expensive. To address this, lifelong knowledge editing techniques (Hartvigsen et al., 2023; Meng et al., 2023; Wang et al., 2024a) have recently emerged. In controlled benchmark studies, they have shown success incorporating new facts along three key metrics: (1) generalization beyond merely memorizing question-answer pairs, (2) locality of edits to maintain pretraining knowledge, and (3) retention of all previously seen edits.

Unfortunately, existing benchmarks are small[2], synthetic, and precede the knowledge cutoff of modern LLMs. While more recent efforts Jang et al. (2022); Khodja et al. (2024) avoid synthetic edits by using wikidata, they remain small-scale (around 20K), and have yet to examine the difficulty of edits, limiting our understanding of *really lifelong* knowledge editing capabilities at practically-needed scales. LLMs are nowadays trained on trillions of text tokens (e.g. Llama-3 (et al., 2024a) on 15T), and studying lifelong factuality updates demands large volumes of recent *real-world* knowledge edits.

To better understand lifelong knowledge editing at a scale and scenario of practical importance, we introduce `WikiBigEdit` - a large-scale, *lifelong* benchmark (Prabhu et al., 2024; Ghosh et al., 2024) with a *fully automated dataset extraction pipeline* that continuously expands with new real-world factual edits to ensure future-proof factuality probing. At its inception, `WikiBigEdit` covers over 500K high-quality question-answer pairs (around 7 million tokens; magnitudes larger than standard edit benchmarks) spanning eight time-intervals over five months (February - July 2024) - reflecting a typical interval between subsequent version releases of large-scale LLMs[3] to better emulate the deployment time required to cover with factual updates.

Derived from periodic changes to Wikidata knowledge graphs (Jang et al., 2022; Khodja et al., 2024), `WikiBigEdit` covers a large range of factual edits and refinements. Moreover, `WikiBigEdit` introduces comprehensive evaluation axes going beyond standard knowl-

---

[1]Tübingen AI Center, University of Tübingen [2]Helmholtz Munich [3]Munich Center for Machine Learning (MCML) [4]Technical University, Munich [5]University of Virginia. Correspondence to: Lukas, Thede <lukas.thede@uni-tuebingen.de>.

*Proceedings of the 42$^{nd}$ International Conference on Machine Learning*, Vancouver, Canada. PMLR 267, 2025. Copyright 2025 by the author(s).

[1]Code available at https://github.com/ExplainableML/WikiBigEdit.

[2]CounterFact (Meng et al., 2023) at 20K, ZsRE (Levy et al., 2017) at 1K, SelfCheckGPT (Manakul et al., 2023) at 600.

[3]E.g. Llama 1 (Touvron et al., 2023) to LLama 2 (et al., 2023), DeepSeek-V1 (et al., 2024b) to V2 (et al., 2024c) and DeepSeek-V3 (et al., 2024d).

edge editing performance measures, incorporating locality checks, multi-hop reasoning across incorporated edits as well as standard and complex generalization tests. We use `WikiBigEdit` to explore the properties of real-world knowledge edits and conduct an extensive study of knowledge editing techniques for `WikiBigEdit`-scale updates, focusing on their ability to integrate numerous sequential edits while maintaining previous knowledge and factual accuracy beyond simple memorization. Additionally, we compare these methods against other generic modification techniques such as retrieval augmentation (Vu et al., 2023; Prabhu et al., 2023a; Gui et al., 2024) and continual fine-tuning (Ibrahim et al., 2024; Roth et al., 2024). Our study reveals limited transfer of knowledge editing techniques to practically scalable lifelong updates and highlights that even when evaluated around knowledge editing desiderata, standard approaches such as retrieval augmentation or continual finetuning with model merging can perform better.

Overall, we make the following contributions to the study of real-world lifelong knowledge editing as scale:

- We present `WikiBigEdit`: a large-scale benchmark of 500k high-quality question-answer pairs spanning eight timesteps over six months, derived from real-world factual changes in Wikidata to both analyze and simulate real-world knowledge edit tasks at a practically relevant scale.

- A fully automated extraction pipeline which continuously extracts new, suitable factual edits from Wikidata knowledge graphs, enabling the benchmark to evolve with new knowledge updates continuously.

- Using `WikiBigEdit`, we thoroughly analyze the capability of existing lifelong knowledge editing methods to conduct lifelong edits at scale; contrasted against retrieval augmentation and continual finetuning to understand limits in relation to other established approaches.

## 2. Related Works

**(Lifelong) Knowledge Editing Benchmarks.** Traditional QA datasets like NQ (Kwiatkowski et al., 2019), TriviaQA (Joshi et al., 2017), and MS MARCO (Bajaj et al., 2018) target open-domain factual QA, HotpotQA (Yang et al., 2018) and EX-FEVER (Ma et al., 2024) assess multi-hop reasoning. Fan et al. (2019); Li et al. (2024); Cheng et al. (2024); Mallen et al. (2023) focus on long-form explanatory questions and time-sensitive QA. StreamingQA (Liška et al., 2022), ArchivalQA (Wang et al., 2021) or HelloFresh (Franzmeyer et al., 2024) address evolving or time-sensitive, event-driven data, while TiC-LM (Li et al., 2025) explores continual pretraining of LLMs. Knowledge editing benchmarks assess the ability to integrate

| Benchmark | Size | Date | Data Source | Task | Lifelong | Mhop |
|---|---|---|---|---|---|---|
| NQ | 307K | 2016 | Google Search queries | Open factual QA | ✗ | ✗ |
| Trivia QA | 650K | Various | Trivia sources (web) | Trivia QA | ✗ | ✗ |
| MS MARCO | 1M+ | 2016 | Bing Search queries | Search queries | ✗ | ✗ |
| Hotpot QA | 112K | 2018 | Wikipedia (curated) | Multi-hop QA | ✗ | ✓ |
| FEVER | 185K | 2018 | Human-written claims | Fact verification | ✗ | ✗ |
| EX-FEVER | 60K | 2023 | Hyperlinked Wikipedia | Mhop fact verif. | ✗ | ✓ |
| ELI5 | 270K | 2019 | Reddit (ELI5 subreddit) | Long-form QA | ✗ | ✗ |
| WikiQA | 3.5K | 2015 | Wikipedia | Factoid QA | ✗ | ✗ |
| DatedData | 200K | Various | Varied web sources | Temporal QA | ✗ | ✗ |
| StreamingQA | 150K | 2007-2020 | WMT news articles | Event-based QA | ✗ | ✗ |
| ArchivalQA | 530K | 1985-2008 | Historical news archives | Fact-based QA | ✗ | ✗ |
| Hello Fresh | 30K | 2023-2024 | X and Wikipedia | Fact verification | ✗ | ✗ |
| CLARK | 1.4K | 2021-2024 | Wikipedia | Knowledge QA | ✗ | ✗ |
| PopQA | 14k | 2023 | Wikipedia, Wikidata | Fact-based QA | ✗ | ✗ |
| TemporalWiki | 7K | 2021 | Wikipedia, Wikidata | Temporal QA | ✓ | ✗ |
| WikiFactDiff | 20k | 2021-2023 | Wikidata | Factual cloze tests | ✗ | ✗ |
| **WikiBigEdit** | **502K** | **2024** | **Wikidata** | **Fact-based QA** | **✓** | **✓** |

*Table 1.* **Comparison of QA and edit benchmarks** in terms of size, date, question types, sources, and lifelong applicability; highlighting the unique placement of our `WikiBigEdit` benchmark.

new facts without retraining. ZsRE (Levy et al., 2017) and COUNTERFACT (Meng et al., 2023) focus on synthetic updates, while AToKE (Yin et al., 2024) and ChronoEdit (Ge et al., 2024) evaluate temporal consistency but remain limited in scale and scope. ELKEN (Peng et al., 2024) and EVEDIT (Liu et al., 2024b) generate edits based on real-world events. TemporalWiki (Jang et al., 2022) and WikiFactDiff[4] (Khodja et al., 2024) are more closely aligned with `WikiBigEdit`. Both evaluate temporal update consistency using consecutive Wikidata snapshots. `WikiBigEdit` provides over a magnitude more samples (Table 2), future-proofing and a comprehensive evaluation along all editing desiderata.

**Knowledge Editing.** Approaches for integrating new knowledge into LLMs include local modification methods such as ROME (Meng et al., 2023) and MEMIT (Meng et al., 2022), global optimization techniques like MEND (Mitchell et al., 2021), and external memorization strategies such as SERAC (Mitchell et al., 2022), GRACE (Hartvigsen et al., 2023), RECIPE (Chen et al., 2024), and WISE (Wang et al., 2024a). While these techniques support efficient model updates, they often suffer from catastrophic forgetting and knowledge degradation when applied to large-scale or sequential edits (Hsueh et al., 2024; Gupta et al., 2024b; Yang et al., 2024b). Our benchmark offers a practical framework to examine these limitations under realistic, evolving knowledge conditions.

**Retrieval Augmented Generation (RAG).** By combining a retrieval module with a generative language model, this class of methods addresses issues such as outdated knowledge and untraceable generations (Lewis et al., 2020; Guu et al., 2020; Karpukhin et al., 2020). RAG approaches have shown strong performance on knowledge-intensive tasks, even at smaller model scales (Borgeaud et al., 2022; Izacard et al., 2022), and enable attribution by linking generated outputs

---

[4]While WikiFactDiff contains over 450K triple–text pairs, only a 20K subset (WFD$_{repl}$) is commonly used for editing due to its defined evaluation protocol.

to retrieved sources (Rashkin et al., 2022; Honovich et al., 2022). Moreover, RAG supports dynamic knowledge updates, mitigating context-memory conflicts when parametric information becomes stale (Vu et al., 2023). However, RAG faces limitations in multi-hop reasoning and compositional retrieval. More advanced variants - such as HippoRAG (Gutiérrez et al., 2025) or agent-based RAG systems - aim to address these limitations by improving reasoning over retrieved information.

**Continual Finetuning and Model Merging.** Continual fine-tuning has emerged as a reliable approach for general large-scale model updates (Garg et al., 2024; Prabhu et al., 2023b; Ibrahim et al., 2024). Roth et al. (2024) show that parameter-efficient finetuning via e.g. LoRA (Hu et al., 2021) or GaLore (Zhao et al., 2024), can be very effective, leveraging regularization provided by limiting trainable parameter counts (Thede et al., 2024). Parameter-selective techniques, including BitFit (Zaken et al., 2022) and LNFit (Min et al., 2023), further reduce computational overhead by targeting specific parameter subsets. Additionally, regularization strategies from continual learning, such as EWC (Kirkpatrick et al., 2017) and SI (Zenke et al., 2017), have shown notable success in mitigating forgetting during the continual adaptation of foundation models. Model merging (Wortsman et al., 2022; Ilharco et al., 2022; Ramé et al., 2024) has shown promising gains coupled with finetuning for continual tasks (Stojanovski et al., 2022; Marouf et al., 2024; Kozal et al., 2024; Marczak et al., 2024; Dziadzio et al., 2024). We investigate and contrast these approaches against knowledge editing at scale.

## 3. `WikiBigEdit`

`WikiBigEdit` addresses three key aspects: (1) it extracts a large number of realistic knowledge edits, (2) incorporates extensive evaluations to assess generalization, reasoning, and locality, and (3) operates fully automatically. The complete pipeline is shown in Figure 1.

**Problem Definition.** The task of large-scale, lifelong knowledge editing requires updating a pre-trained LLM $f^0$ with a stream of factual updates $[u_1, u_2, \ldots, u_T]$, where each update $u_t = (q_t, a_t)$ consists of a question $q_t$ and its factual answer $a_t$. These updates are grouped into $B$ sequential batches $[U_1, U_2, \ldots, U_B]$, where each batch can encompass anything from a single edit to multiple. For each batch update $b$ (also denoted as *timestep* in this work), the model $f^{b-1}$, trained on updates from prior batches $U_{<b}$, is further updated with the current batch $U_b$ to produce $f^b$. The model must meet three key objectives: (1) integrate updates from $U_b$, (2) retain knowledge from $U_{<b}$, and (3) ensure that the semantic understanding required to correctly solve previously seen edits generalizes beyond simple memorization

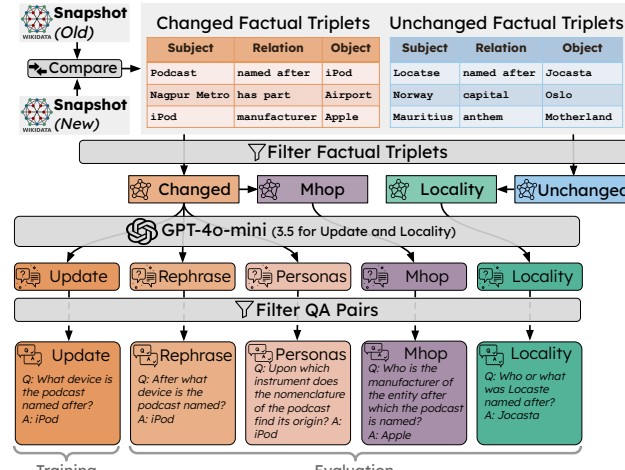

*Figure 1.* **Automated Dataset Generation Pipeline.** (1) Comparing two Wikidata knowledge graph snapshots; extract sets of unchanged and changed subject-relation-object triplets, (2) filter for quality, (3) generate locality probes, (4) produce multi-hop reasoning qintuples, (5) convert into question-answer pairs using e.g. GPT-3.5, (6) introduce rephrased and persona-based evaluations to further test generalization, conduct (7) final quality assurance. Put together, this can continuously extend `WikiBigEdit` with additional time-specific high-quality knowledge edit data.

of exact question-answer pairings.

### 3.1. Lifelong Automated Extraction of Factual Edits

As a proxy for world knowledge, we leverage Wikidata knowledge graphs, following recent works (Jang et al., 2022; Khodja et al., 2024). By performing pairwise comparisons of snapshots taken at different points in time, we identify changes that approximate real-world knowledge evolution. We thus automatically generate `WikiBigEdit` using our proposed seven-step pipeline, as follows:

**(1) Periodic Snapshot Acquisition.** The automated pipeline is initiated through periodic downloads of most recent snapshots of the Wikidata knowledge graph, and paired with the last downloaded snapshot. Pairwise comparison of these snapshots allows us to extract two distinct sets of *subject-relation-object triplets* $\mathcal{T} = (\text{subj}, \text{rel}, \text{obj})$, $\mathcal{S} = \{\mathcal{T}_i\}$: $\mathcal{S}_{\text{static}}$ representing *unchanged facts* and $\mathcal{S}_{\text{changed}}$ capturing *updated or newly added information* (c.f. top of Figure 1). While $\mathcal{S}_{\text{changed}}$ will be used to create the knowledge edits to be incorporated into a base model, $\mathcal{S}_{\text{fixed}}$ is primarily utilized for evaluations. Each triplet describes a given relation (such as "*capital*", "*manufacturer*", ...) between a subject (e.g. "*Norway*") and object (s.a. "*Oslo*"). A detailed overview is provided in the supplementary.

**(2) Initial Filtering.** To ensure high-quality triplets, we apply several filtering steps to both $\mathcal{S}_{\text{static}}$ and $\mathcal{S}_{\text{changed}}$: (*i*) We exclude triplets with circular dependencies, (*ii*) where the

subject or object entities consist of non-Roman characters, (*iii*) single characters, (*iv*) overly long phrases (e.g., more than five words) and (*v*) where a single subject-relation combination maps to multiple objects to avoid ambiguity.

**(3) Generation of Locality Probes.** The filtered $\mathcal{S}_{\text{changed}}$ defines the knowledge updates. To evaluate whether knowledge edits remain *local*, i.e. do not influence (related) existing knowledge, we derive a *locality* set $\mathcal{S}_{\text{locality}}$ from $\mathcal{S}_{\text{static}}$ and $\mathcal{S}_{\text{changed}}$; where each factual triplet in the changed set is paired with a triplet *semantically similar* in subject and relation but with a distinct object. For each factual update, this gives a corresponding question whose answer should remain unaffected to probe the locality of knowledge edits.

**(4) Inclusion of Multi-Hop Reasoning Tuples.** To enable reasoning evaluation, we further extract multi-hop (**mhop**) tuples $\mathcal{S}_{\text{mhop}}$ from $\mathcal{S}_{\text{changed}}$. In particular, we identify pairs of factual triplets $(\mathcal{T}_i, \mathcal{T}_j)$ where $\mathcal{T}_i^{\text{obj}}$ serves as the subject of the second triplet, $\mathcal{T}_j^{\text{subj}}$; for example $\mathcal{T}_i = $ (*podcast, named after, iPod*) and $\mathcal{T}_j = $ (*iPod, manufacturer, Apple*). The triplets $\mathcal{T}_i$ and $\mathcal{T}_j$ are then concatenated to form two-hop factual qintuples $\mathcal{T}_{ij}$, e.g. (*podcast, named after, iPod, manufacturer, Apple*). By restricting the extraction of $\mathcal{S}_{\text{mhop}}$ to $\mathcal{S}_{\text{changed}}$, we ensure a fair evaluation, as the model will have *been updated with both individual facts* prior to the reasoning task. We find that for a filtered batch $\mathcal{S}_{\text{changed}}$, we can recombine around $3.9\%$ update triplets into $\mathcal{S}_{\text{mhop}}$.

**(5) Generation of Question-Answer Edit Pairs.** For $\mathcal{S}_{\text{changed}}$, $\mathcal{S}_{\text{locality}}$, and $\mathcal{S}_{\text{mhop}}$, we generate corresponding question-answer pairs using GPT-4o-mini (Brown et al., 2020) (GPT-3.5 for update and locality) through few-shot prompting (for all prompt templates, please see the supplementary). For $\mathcal{S}_{\text{mhop}}$, we generate questions that omit the middle entity of the two-hop factual quintuple. Using this, we generate the corresponding question-answer sets $\mathcal{S}_{\text{changed}}^{\text{QA}}$, $\mathcal{S}_{\text{locality}}^{\text{QA}}$, $\mathcal{S}_{\text{mhop}}^{\text{QA}}$. $\mathcal{S}_{\text{locality}}^{\text{QA}}$ and $\mathcal{S}_{\text{mhop}}^{\text{QA}}$ are used for evaluation, while $\mathcal{S}_{\text{changed}}^{\text{QA}}$ constitutes the respective fact-based training data.

**(6) Generation of Personalized and Rephrased Question-Answer Edit Pairs for Generalization Studies.** While $\mathcal{S}_{\text{locality}}^{\text{QA}}$ and $\mathcal{S}_{\text{mhop}}^{\text{QA}}$ allow us to probe locality and reasoning generalization of edits, we also provide $\mathcal{S}_{\text{rephrase}}^{\text{QA}}$ and $\mathcal{S}_{\text{personas}}^{\text{QA}}$ containing rephrased variants of $\mathcal{S}_{\text{changed}}^{\text{QA}}$ to check generalization of edits beyond memorization. In particular, we utilize GPT-4o-mini to provide additional question-answer variations of $\mathcal{S}_{\text{changed}}^{\text{QA}}$ by enforcing the language model to either directly rephrase $\mathcal{S}_{\text{changed}}^{\text{QA}}$ to produce $\mathcal{S}_{\text{rephrased}}^{\text{QA}}$ or directly generate $\mathcal{S}_{\text{personas}}^{\text{QA}}$ by mimicking various different "*personas*" (e.g. "*detective*", "*pirate*", "*philosopher*" defined by a detailed description, see supplementary). For both $\mathcal{S}_{\text{rephrased}}^{\text{QA}}$ and $\mathcal{S}_{\text{personas}}^{\text{QA}}$, post-hoc filtering is applied to ensure that sub-

| Timestep | Date Range | Samples | Unsolved |
|----------|------------|---------|----------|
| T1 | 2024/02/01 - 2024/02/20 | 26,922 | 80% |
| T2 | 2024/02/20 - 2024/03/01 | 29,835 | 84% |
| T3 | 2024/03/01 - 2024/03/20 | 54,504 | 85% |
| T4 | 2024/03/20 - 2024/04/01 | 43,443 | 85% |
| T5 | 2024/04/01 - 2024/05/01 | 121,116 | 81% |
| T6 | 2024/05/01 - 2024/06/01 | 101,728 | 82% |
| T7 | 2024/06/01 - 2024/06/20 | 69,403 | 82% |
| T8 | 2024/06/20 - 2024/07/01 | 55,431 | 82% |
| | **Total** | **502,382** | **82%** |

*Table 2.* **Description of `WikiBigEdit` timesteps.** Sample counts reflect number of question-answer pairs associated with a given time-interval; alongside information highlighting the percentage of unsolved edits across studied LLMs; highlighting the large number of required real-world factual updates across LLMs.

ject, object and relation remain correctly encoded in both.

**(7) Final Filtering Stage.** A final filtering process ensures the quality of the generated question-answer pairs for both training and evaluation sets by verifying that the generated questions contain the respective subject and relation *while not giving away the answer* and answers incorporate the right object. After this stage, the final $\mathcal{S}_{\text{changed}}^{\text{QA}}$, $\mathcal{S}_{\text{locality}}^{\text{QA}}$, $\mathcal{S}_{\text{mhop}}^{\text{QA}}$, $\mathcal{S}_{\text{rephrased}}^{\text{QA}}$ an $\mathcal{S}_{\text{personas}}^{\text{QA}}$ are incorporated as a new knowledge edit batch into `WikiBigEdit`.

### 3.2. Analysis and Exploration

In this section, we first summarize various benchmark characteristics before leveraging the first instantiation of our large-scale `WikiBigEdit` to study the structure and properties of real-world knowledge edits derived from Wikidata knowledge graphs. Table 2 provides an overview of the *eight time-intervals* that constitute `WikiBigEdit`. The factual updates are extracted on a roughly bi-monthly frequency from Wikidata snapshots spanning February to July 2024; with the availability of Wikidata snapshots determining size and extend of factual updates associated with each timestep.

To ensure high data quality, we filter out approximately 76% of initial changes, removing cyclic or overly long entities (11%), non-entity-to-entity relations (50%), and nondeterministic triplets (15%). The filtered updates range from 27k to 69k per step, with monthly factual changes exceeding 100k. This variability reflects the dynamic nature of knowledge updates and practical constraints when aiming to ensure that models remain up-to-date. Overall, this version contains 502,382 question-answer pairs across 6.9M tokens.

**Impact of Language Model Choice on QA Generation** The quality of generated question-answer pairs in `WikiBigEdit` depends in part on the underlying language model used during generation. Before we conduct further experiments, we begin by examining the sensitivity of the generation process to different LLMs by comparing out-

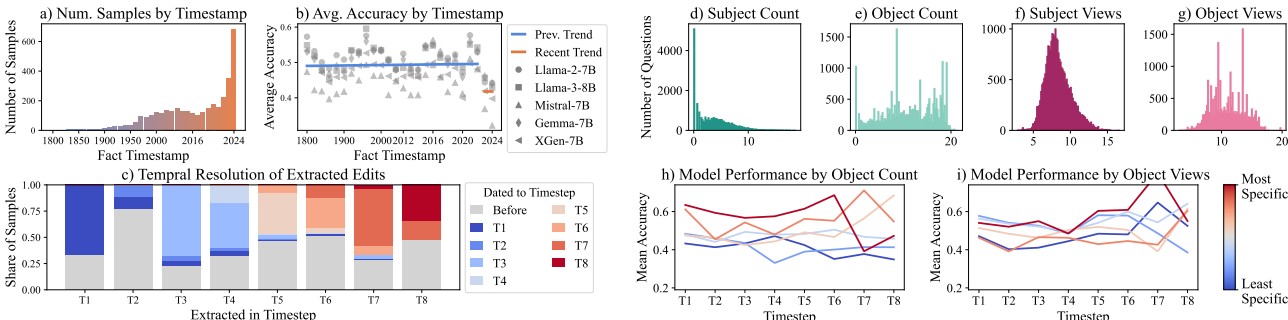

*Figure 2.* **Analysis and Exploration of Real-World Factual Updates in `WikiBigEdit`.** **(a)** Distribution of time-stamps associated with factoids where temporal information was available. For real-world updates, facts cover both historical, but especially more recent context. **(b)** Average accuracy of five LLMs evaluated across facts broken down by timestamps. We find that contrasted against facts clearly preceding knowledge cutoffs (pre-2023), there is a notable drop in average capabilities, highlighting the need to both address edits on pretraining context, but especially the inclusion of recent facts or corrections. **(c)** Distribution of edits extracted for each timestep and their corresponding dates. While the majority of edits align with their respective timestep, edits in larger parts also incorporate changes of information associated with primarily "*older*" time intervals. **(d-g)** Real-world edits cover both highly specific facts (measured on subject and object counts in pretraining corpora) as well as general, common knowledge modifications. **(h-i)** Especially on highly specific edits as defined by pretraining concept frequency does performance go down.

| *Update Triplet* | *Lea County Regional Airport - state of use - in use* |
| --- | --- |
| **GPT-3.5** | What is the current state of use for Lea County Regional Airport? |
| **GPT-4o-mini** | What is the current state of use of Lea County Regional Airport? |
| **GPT-4o** | What is the current state of use of Lea County Regional Airport? |
| *Mhop Tuple* | *2004 VS75 - discoverer - Marc Buie - gender - male* |
| **GPT-3.5** | Who is the discoverer or inventor of 2004 VS75? |
| **GPT-4o-mini** | What is the sex or gender of the discoverer of 2004 VS75? |
| **GPT-4o** | What is the gender of the discoverer or inventor of 2004 VS75? |

*Table 3.* Comparison of generated questions across different language models for a simple factual update (top) and a multi-hop reasoning example (bottom). While all models perform well on the single-hop case, GPT-3.5 struggles with compositional reasoning, often producing incomplete multi-hop questions. GPT-4o-mini and GPT-4o generate more complete and semantically accurate queries.

puts from GPT-3.5-turbo, GPT-4o-mini, and GPT-4o. For both the *Update* and *Locality* sets, which are directly derived from structured subject–relation–object triplets, model choice has negligible impact. Even the less capable GPT-3.5-turbo consistently produced well-formed, semantically correct questions. As shown in Table 3, the outputs across all models for such simple factual prompts are stylistically comparable and accurate. In contrast, the *Multi-hop* and *Persona* sets were more sensitive to model choice, as they require compositional reasoning and stylistic transformation. For instance, GPT-3.5-turbo often struggled to combine facts effectively in multi-hop settings, producing incomplete or ambiguous questions. GPT-4o-mini and GPT-4o, in comparison, reliably produced complete and coherent outputs. These observations confirm our selection for GPT-4o-mini to generate *Multi-hop* and *Persona* sets due to its strong balance between generation quality and computational efficiency.

**Understanding the temporal resolution of real-world factual updates.** To assess the temporal resolution of knowledge edits in our benchmark, we analyze timestamps for factual triplets where date information is available (10,409 question-answer pairs). As shown in Figure 2 (**a**), updates strongly focus on recent events, though many also modify facts predating the edit dates. Figure 2 (**c**) further reveals that extracted updates are not strictly confined to their respective timesteps, underscoring the dual challenge of integrating both new knowledge and modifying existing facts from pretraining corpora. When stratified by knowledge cutoff (pre-2023 vs. 2023/2024), all tested LLMs perform worse on post-cutoff facts (Cheng et al., 2024) (**b**), highlighting the critical need for effective integration of new knowledge.

**Specificity of factual updates.** We assess the specificity of knowledge edits by distinguishing well-known from highly specialized facts, using two measures: (1) entity frequency from the Infinigram API (Liu et al., 2024a), and (2) Wikipedia view counts (Jan 2023–June 2024) (Mallen et al., 2023). As shown in Figure 2 (**d–g**), `WikiBigEdit` includes both general (e.g., *Which continent is Krasnodar a part of?*) and highly specific questions (e.g., *Who was the doctoral advisor of Omar Khayyám?*). To evaluate how specificity affects performance, we test five LLMs (Llama-2-7b, Llama-3-8b, Mistral-7b, Gemma-7b, xGen-7b) and correlate their accuracy. As shown in (**h**), lower entity frequency correlates with reduced performance, consistent with Udandarao et al. (2024); Cheng et al. (2024), while view counts (**i**) show less consistent trends. Specificity appears primarily driven by the object rather than the subject. See supplementary for details.

**WikiBigEdit - Analysis Summary**

When evaluating LLMs on a large number of real-world factual updates, performance is at best mediocre, but especially low for highly specific and recent facts. For LLMs to serve as accurate and timely knowledge bases, the ability to edit knowledge across historic, recent or specific context at scale is thus crucial.

## 4. Understanding the Limits of Lifelong Knowledge Editing in LLMs

Leveraging `WikiBigEdit`, we evaluate the ability of lifelong knowledge editing techniques to incorporate large-scale factual updates. Section 4.1 presents a comprehensive application of various lifelong editing approaches and compares them to alternative methods like continual finetuning and retrieval augmentation. Section 4.2 analyzes the specific limitations of knowledge editing techniques, with Section 4.3 and Section 4.4 providing detailed evaluations of retrieval augmentation and continual finetuning.

**Experimental Details.** All experiments are performed on a compute cluster equipped with Nvidia A100 and H100 GPUs, leveraging PyTorch (Paszke et al., 2019) and building on the EasyEdit codebase (Zhang et al., 2024). All our evaluations are conducted on five state-of-the-art LLMs: Llama-2-7b (et al., 2023) Llama-3-8b (et al., 2024a), Mistral-7b (Jiang et al., 2023), Gemma-7b (Team et al., 2024a) and xGen-7b (Nijkamp et al., 2023). To representatively study knowledge editing, we leverage the following approaches: ROME (Meng et al., 2023), R-ROME (Gupta et al., 2024a), MEMIT (Meng et al., 2022), and WISE (Wang et al., 2024a). For our retrieval-augmented generation experiments, we store update question-answer pairs in memory alongside associated key embeddings generated by embedding questions with the *all-mpnet-base-v2* embedding model (Song et al., 2020), the best general sentence-embedding model provided in the SBERT repository (Reimers & Gurevych, 2019). During inference, the top-2 most similar entries are retrieved from memory efficiently using the Annoy solver (Bernhardsson, 2018). This setup ensures efficient retrieval and supports multi-hop questions with minimal computational overhead. For our continual finetuning experiments, we explore variations of low-rank adaptation using LoRA (Hu et al., 2021) alongside simple interpolation-based model-merging (Wortsman et al., 2022; Stojanovski et al., 2022; Marouf et al., 2024) which has shown to perform competitively even when compared to more involved merging techniques (Roth et al., 2024; Dziadzio et al., 2024). Adapters are trained for 10 epochs on each timestep. Additional details are provided in the corresponding subsections and the supplementary.

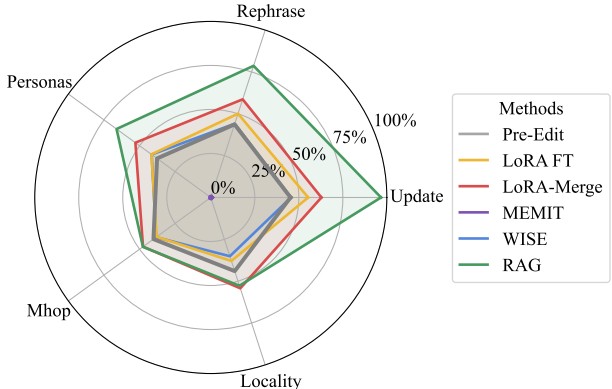

*Figure 3.* **Large-Scale Lifelong Knowledge Editing on `WikiBigEdit`.** Comparison of knowledge editing techniques and other standards for model modification (retrieval-augmented generation, RAG, and continual finetuning using LoRA + merging). Results reveal that RAG vastly outperforms specialized knowledge editing techniques (at higher inference cost). At equivalent inference, simple continual finetuning consistently improves on advanced editing techniques at scale.

### 4.1. Lifelong Knowledge Editing on `WikiBigEdit`

To begin, Figure 3 presents a high-level comparison of the average accuracy across all eight timesteps $[U_1, ..., U_8]$ for all five LLMs listed above. After incorporating factual changes of a timestep, evaluation is conducted across all five evaluation axes spanned by respective evaluation data $\mathcal{S}^{QA}_{rephrased}$, $\mathcal{S}^{QA}_{personas}$, $\mathcal{S}^{QA}_{locality}$, $\mathcal{S}^{QA}_{mhop}$, and a performance evaluation on the actual edit data $\mathcal{S}^{QA}_{update}$. Results across all models and timesteps are averaged and visualized.

RAG (green line) proves to be the most effective approach, significantly outperforming knowledge editing techniques across evaluation axes, nearly tripling accuracy on training edits. This result is expected, as memorizing previously seen edits makes solving the task straightforward. Memorization also boosts performance on generalization and locality metrics, commonly used in knowledge editing literature. However, compositional reasoning on multihop questions sees only slight improvements, as expected.

RAG naturally trades performance increases for longer inference times. However, with effective approximate solvers and high-quality sentence embeddings, we find that inference time at most doubles after processing all of `WikiBigEdit` (c.f. Figure 16). Even so, knowledge editing techniques like WISE and MEMIT, specifically designed to factually update LLMs, cannot match simple low-rank adapter-based finetuning (*LoRA-FT*) at scale. Moreover, with interpolation-based model merging (*LoRA-Merge*), commonly used in training and finetuning LLMs (Yang et al., 2024a; Roth et al., 2024; Team et al., 2024a;b; Dziadzio et al., 2024), consistent gains are observed across

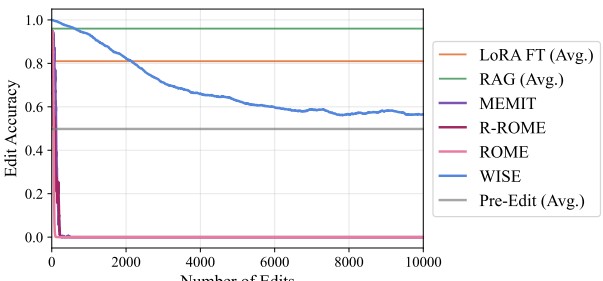

*Figure 4.* **Performance Decay of Knowledge Editing Techniques.** Update accuracy over first 10K edits for knowledge editing methods compared to simple baselines that show higher performance (LoRA, RAG). Local modification methods collapse early (MEMIT, ROME, R-ROME), and even those specifically designed for large volumes of edits (WISE) converge to pre-edit accuracy.

*all* evaluation axes. These results strongly suggest that at scale, standard knowledge editing techniques fall short even when compared to straightforward continual finetuning approaches commonly employed for general continual pretraining (Ibrahim et al., 2024; Roth et al., 2024).

> **Key Takeaway**
>
> Updated via `WikiBigEdit`, RAG significantly outperforms all approaches at double the inference time; though expectedly struggles with multi-hop reasoning. *Given a fixed inference budget*, we find that at scale, knowledge editing approaches evaluated across all standard editing evaluation axes are actually matched or outperformed by simple continual finetuning, especially using merging.

The following sections provide a more detailed breakdown of specific failure scenarios and drivers.

### 4.2. The Limits of Knowledge Editing at Scale

This section provides an in-depth analysis of knowledge editing applied to long and realistic update sequences, with results shown in Figure 4. Using Llama-2-7b as the underlying LLM, we present the first 10K update steps from the initial timestep in `WikiBigEdit` for clarity. A complete version with all models is available in the supplementary.

The accuracy of ROME, R-ROME, and MEMIT - local knowledge editing techniques that locally modify specific model weights - rapidly degrades within the first few hundred updates, leading to model collapse. This failure on large-scale, sequential knowledge editing aligns with recent findings (Gupta et al., 2024b; Yang et al., 2024b) and highlights difficulties for lifelong knowledge editing at scale. Even without direct model edits, methods like WISE - designed for larger update volumes by storing adapted weight matrices for retrieval during inference - show limited bene-

fits. While WISE avoids catastrophic failure and integrates new facts over longer update sequences, its performance steadily declines within the first 10K updates, ultimately *converging to pre-update knowledge levels*. This suggests that by avoiding direct model modifications, WISE struggles to retain updates over time, making it unsuitable for large-scale sequential integration, especially compared to retrieval augmentation and simple continual fine-tuning, which incur no additional inference cost.

> **Key Takeaway**
>
> Knowledge editing is reliant on local weight modification and collapses quickly. Even methods designed for lifelong editing converge to pre-update performance early. So large-scale lifelong editing requires significantly improving existing methods.

### 4.3. Retrieval Augmentation for `WikiBigEdit`

As shown in Figure 3, retrieval-augmented generation (RAG) demonstrates significant improvements over direct knowledge editing techniques. While expected for performance on training edits, the increased performance across all evaluation axes, including generality, is somewhat surprising, as prior retrieval approaches like GRACE (Hartvigsen et al., 2023) have shown limited generalization capabilities (Wang et al., 2024a). However, on realistic edits and at scale, we find that efficiently implemented retrieval-augmentation outperforms any editing method (see Section 4 and supplementary for details).

Looking at the detailed results of the RAG baseline on `WikiBigEdit` in Figure 5, the improvements across all evaluation axes (except multi-hop reasoning) are significant across 500K+ updates. The top row shows the average accuracy on the current timestep, with bold lines representing average performance across five models (light background lines). The bottom row illustrates average performance across all timesteps after training on the current timestep's edits. As noted previously, RAG achieved near-perfect accuracy on $\mathcal{S}_{update}^{QA}$ by simply retrieving the exact stored questions from memory; exhibiting no sign of "*forgetting*" on trained edits (c.f. bottom row, leftmost). Additionally, edits remain highly localized since no weight updates are performed. Interestingly, while RAG also maintains high average performance on generalization-based evaluation axes ("*rephrasing*" and "*personas*"), achieving average accuracies of 78.64% and 66.16%, respectively, RAG still *exhibits symptoms akin to catastrophic forgetting* encountered in continual learning and knowledge editing (Kirkpatrick et al., 2017; Zenke et al., 2017; Wang et al., 2024a). Here, performance on older timestep data decays with every new large update batch (bottom, second and third), as the retrieval

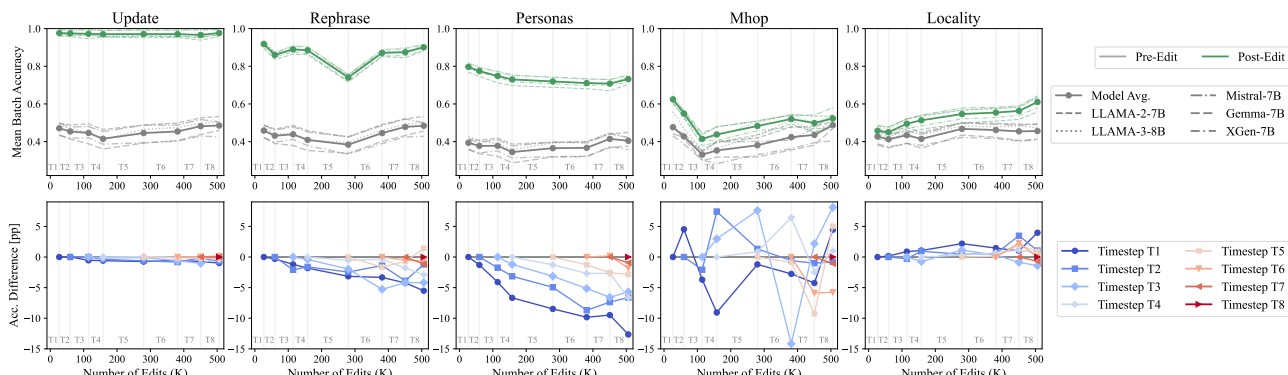

*Figure 5.* **Retrieval-Augmented Generation for Lifelong Knowledge Editing**. **(Top)** Mean batch accuracy for all evaluations: update, rephrase, personas, multi-hop and locality across all models. **(Bottom)** Accuracy changes for preceding timesteps after training on a given timestep, highlighting forgetting symptoms on generalization tasks, and marginal benefits at best for reasoning over edits.

memory becomes more and more obfuscated with context irrelevant for data from particular timesteps.

Finally, while RAG outperforms knowledge editing approaches on standard measures (locality, generalization, and overall edit performance), its multi-hop reasoning accuracy remains near pre-update levels (second from right in Figure 5). Figure 6 reveals the bottleneck: retrieval accuracy is low - likely due to the absence of the middle subject in the multi-hop question - with correct retrieval rates below 50% for the first hop and even lower for the second. Only 10% of evaluation samples retrieve both facts correctly. Even when both hops are retrieved, answer accuracy remains near single-hop levels (i.e. solving multi-hop questions using only information from just the first hop), as models struggle to combine retrieved information without dedicated training for reasoning tasks. While advanced methods like HippoRAG (Gutiérrez et al., 2025) may help, they typically require longer contexts or higher compute; we leave their integration to future work.

> **Key Takeaway**
>
> Simple RAG outperforms much more involved knowledge editing techniques across all editing evaluation axes at larger scale and real-world edits. However, performance on evaluation over older edits still decays, and reasoning over edited facts shows marginal benefits at best.

### 4.4. Continual Finetuning

To conclude, we investigate continual finetuning using low-rank adaptation (LoRA) as a simple yet effective method to incorporate large volumes of new facts. While recent works (e.g., Thede et al. (2024); Roth et al. (2024)) have demonstrated the benefits of continual low-rank finetuning for general continual learning tasks, we show that these ad-

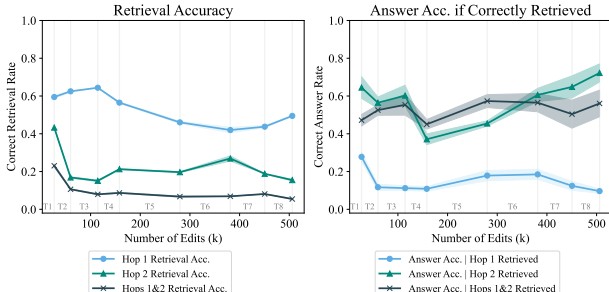

*Figure 6.* **Multi-hop RAG. (Left)** Retrieval accuracy for individual and both hops, highlighting significant challenges in retrieving both relevant facts simultaneously. **(Right)** Answer accuracy conditioned on correct retrieval, showing that even when both hops are retrieved, models struggle to combine the information.

vantages extend to specialized knowledge editing for LLMs. As shown in Figure 3, simple non-specialized finetuning over all `WikiBigEdit` updates outperforms specialized knowledge editing techniques across all standard evaluation axes, with gains further amplified by interpolation-based model merging. These findings emphasize the need to revisit design choices in knowledge editing, evaluate on larger, realistic benchmarks, and improve baseline comparisons.

Finally, we provide a detailed perspective on lifelong knowledge editing using continual finetuning in Figure 7 (rank-4 LoRA; additional results in the supplementary[5]). As shown in the top row, performance remains competitive in early timesteps but steadily declines as updates accumulate, eventually converging to or even falling below pre-update accuracy, particularly for locality and multi-hop reasoning. This aligns with expectations, as catastrophic forgetting persists even in low-rank adaptation (Thede et al., 2024; Roth et al., 2024). However, for the first 100K updates, continual finetuning shows notable success in integrating edits.

---

[5]High-rank configurations exhibit significant instability, including catastrophic failure on Mistral and poor performance across other models.

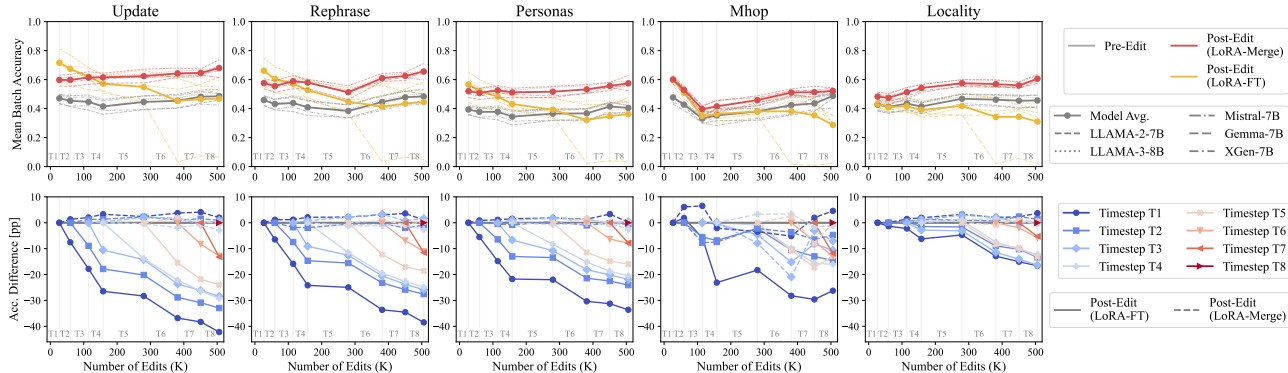

*Figure 7.* **Continual Finetuning for Knowledge Editing. (Top)** Mean batch accuracy for update, rephrase, personas, mhop, and locality sets for continual LoRA finetuning (LoRA-FT) with model merging (LoRA-Merge). **(Bottom)** Accuracy differences for individual timesteps. LoRA-FT degrades over time, while merging ensures consistent editability across all `WikiBigEdit`.

These results highlight its viability for mid-term knowledge integration but also its limitations at scale.

Interestingly however, as we couple continual LoRA finetuning with simple model merging (after training of each timestep, current adapter weights are simply merged into preceding adapter weights using an interpolation weight of 0.25), we achieve significantly improved editing performance across `WikiBigEdit`. While LoRA-Merge initially shows lower performance on shorter update sequences, it demonstrates superior stability as updates accumulate; and begins to notably outperform simple continual finetuning (and thereby also any specialized knowledge editing technique) after approximately 100K edits on all evaluation sets, maintaining *consistent performance without significant degradation*. Importantly, LoRA-Merge shows at most marginal evidence of catastrophic forgetting, and in parts even exhibits positive backward transfer (where training on a current timestep actually improves performance on all edits seen previously) - see Figure 7 bottom row. These results reinforce the effectiveness of model merging for general continual finetuning and pretraining tasks (Stojanovski et al., 2022; Thede et al., 2024; Roth et al., 2024), highlighting its promise for large-scale knowledge editing.

> ### Key Takeaway
>
> Generic adapter-based finetuning matches specialized knowledge editing for mid-level knowledge editing; though exhibits catastrophic forgetting at scale. However, equipped with simple model merging, we find consistent inclusion of new facts across all of `WikiBigEdit`. This suggests better baseline comparisons and studies at scale for the development of specialized knowledge editing techniques.

## 5. Conclusion

This paper explores lifelong knowledge editing for large language models at a practical scale. We introduce `WikiBigEdit`, a benchmark with an automated data acquisition pipeline that extracts real-world factual updates from Wikidata, enabling *lifelong* expansion. In its first version, `WikiBigEdit` contains over 500K QA pairs for large-scale lifelong knowledge editing, complemented by a comprehensive evaluation suite covering standard knowledge editing criteria, reasoning, and generalization.

Leveraging `WikiBigEdit`, we conduct an extensive experimental study in which we identify key limitations of existing knowledge editing approaches such as ROME, MEMIT, and WISE, which struggle with scalability and stability in large-scale update settings. Our results also demonstrate the efficacy of retrieval-augmented generation (RAG) when no restriction on inference time is given and highlight continual finetuning coupled with model merging as a competitive alternative at an equivalent inference cost.

We believe that `WikiBigEdit` provides an important reference to understand the transfer of knowledge editing methods to real-world scale and facts, as well as suggestions towards more complete baseline studies during development of such specialized techniques.

## Acknowledgements

LT and KR thank the International Max Planck Research School for Intelligent Systems (IMPRS-IS) for support. KR also thanks the European Laboratory for Learning and Intelligent Systems (ELLIS) PhD program for support. We are grateful for support by the Carl Zeiss Foundation, project "Certification and Foundations of Safe Machine Learning Systems in Healthcare". This work was partially funded by the ERC (853489 - DEXIM) and the Alfried Krupp von Bohlen und Halbach Foundation, which we thank for their generous support.

## Impact Statement

Maintaining factual accuracy in large language models (LLMs) is critical, as these models are increasingly relied upon in domains where misinformation or outdated knowledge can have serious societal consequences. By focusing on the lifelong integration of factual knowledge, this paper directly addresses the challenge of ensuring that LLMs remain up-to-date, reliable, and accurate over time. The proposed `WikiBigEdit` benchmark provides a valuable tool for designing and evaluating methods that enhance factuality, offering a scalable and realistic framework grounded in real-world knowledge evolution. By advancing techniques for continual factual updates, this work contributes to reducing the risks of outdated or incorrect information, promoting trust and accountability in the use of LLMs for applications in education, healthcare, journalism, and beyond.

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

# A. Benchmark Generation

This section provides additional details about the benchmark generation process, focusing on the templates used to create question-answer pairs from the extracted factual updates. Each subset of the benchmark — update, locality, rephrase, personas, and multi-hop — is generated using carefully designed prompts tailored to their specific evaluation objectives. Below, we describe the prompt templates and methodologies for generating these subsets.

## A.1. Update and Locality Set

The update set serves as the foundation of our benchmark, containing questions derived directly from the extracted factual updates. The locality set, on the other hand, evaluates the robustness of knowledge updates by ensuring that unrelated but similar facts remain unaffected.

We employ the following prompt template to generate both the update and locality sets. The prompt includes a general task description, detailed instructions, and few-shot examples to guide the model in generating high-quality question-answer pairs:

---

**Prompt Template Update/Locality**

Given a fact triplet in the format (subject, relation, object) and a description of the relation, generate a question-answer pair where the answer is the object of the triplet. Ensure that the question explicitly references the relation and is structured to lead uniquely to the chosen answer, which must be the exact text of the object. Use the description of the relation to improve the specificity and clarity of the question. Make the question specific enough to ensure that there is only one possible correct answer, eliminating any ambiguity.

**Instructions for Generation:**

1. Construct the Question:

    - Ensure the question is directly tied to the relation and points exclusively to the selected answer.
    - Include additional context from the subject to make the question more specific and unambiguous.
    - Use the description of the relation to enhance the quality of the question.
    - Use only the exact text of the object in the answer.

2. Format: Provide the question-answer pair in the following format:

    - Q:
    - A:

**Examples:**
Fact Triplet: Turnberry Lighthouse, color, white
Relation Description: The color of the subject.
Q: What is the color of Turnberry Lighthouse?
A: white

Fact Triplet: Folsom Library, part of, Rensselaer Libraries
Relation Description: part of this subject; inverse property of "part of" (P361). See also "has parts of the class" (P2670).
Q: What institution does Folsom Library belong to?
A: Rensselaer Libraries

Fact Triplet: Volt Europa, subsidiary, Volt Netherlands
Relation Description: A company controlled by another company.
Q: What is the name of the subsidiary of Volt Europa in the Netherlands?
A: Volt Netherlands

Fact Triplet: Zonia Baber, educated at, Chicago State University
Relation Description: The institution where the subject received their education.
Q: At which university did Zonia Baber receive her education?
A: Chicago State University

**Task:**
According to the instructions and examples, provide a QA pair for the following fact triplet:

Fact Triplet: {}, {}, {}
Relation Description: {}

---

## A.2. Rephrase Set

The rephrase set is designed to test the generalization of updated knowledge by reformulating the questions from the update set. The reformulated questions retain the same factual context but vary in phrasing, introducing linguistic diversity to the benchmark.

The following prompt is used to generate the rephrased questions:

---

**Prompt Template Rephrase**

You are given a question-answer pair. Reformulate the question such that its content remains identical and the given answer is still the only accurate answer.

Instructions for Generation:

1. Question Reformulation:

   - Restructure the question while maintaining the same content and context.
   - Ensure that the answer provided in the original question remains the only correct answer.

2. Response Format: Provide the reformulated question in the following format:

   - Reformulated Question:

**Examples:** Original Question: What is the capital of the United States?
Answer: Washington, D.C.
Reformulated Question: What city serves as the capital of the United States?

Original Question: Who painted the Mona Lisa?
Answer: Leonardo da Vinci
Reformulated Question: By whom was the Mona Lisa painted?

Original Question: When was the Declaration of Independence signed?
Answer: July 4, 1776
Reformulated Question: On what date was the Declaration of Independence signed?

**Task:** Original Question: {} Answer: {}

---

### A.3. Persona Set

The persona set adds a layer of complexity by requiring the model to generate questions in the style of distinct personas. Each question from the update set is rewritten based on the characteristics of a randomly selected persona.

| Persona | Description |
| --- | --- |
| **Detective** | You are a world-weary detective narrating your investigations. Your tone is gritty, mysterious, and evocative of noir fiction. Reformulate the following questions as if you're puzzling out clues in a case. Keep the core meaning intact but phrase it in your distinct detective style. |
| **Casual** | You are a casual, friendly conversationalist with a relaxed and approachable tone. You're curious and engaging, as if you're chatting with a friend over coffee or in a group discussion. Reformulate the following questions to sound natural and conversational while keeping the core meaning intact. |
| **Pirate** | You are a swashbuckling pirate with a flair for colorful language and sea-themed metaphors. Your tone is bold, rough, and full of pirate lingo, as if you're recounting tales from the high seas. Reformulate the following questions to sound like they're being asked by a true pirate. Keep the core meaning intact but add a piratical twist. |
| **Philosopher** | You are an ancient philosopher, deeply contemplative and wise, always pondering the greater truths of existence. Your tone is reflective, profound, and formal, as if you are crafting a dialogue or treatise on the nature of knowledge and events. Reformulate the following questions as if they are inquiries befitting philosophical discourse. Keep the core meaning intact but phrase it in your distinct, thoughtful style. |

| **Caveman** | You are a caveman, speaking in simple and primitive language, with short sentences and limited vocabulary. Your tone is direct, straightforward, and reflects the early stages of human communication. Reformulate the following questions in a way that matches your basic and minimalistic speech style, while keeping the core meaning intact. |
|---|---|

Table 4: Descriptions of the personas used for rephrasing questions with varied styles.

The generation process uses the same structure as the rephrase prompt, with the addition of persona-specific descriptions. Table A.3 provides an overview of the personas used, including a detective, casual speaker, pirate, philosopher, and stone-age character. For each question, one persona is chosen, and its description and few-shot examples are inserted into the rephrase prompt to guide the generation process.

### A.4. Mhop Set

The multi-hop set evaluates the reasoning capabilities of the model by requiring it to combine information from two factual triplets. The questions are designed to omit the middle entity, prompting the model to infer the correct answer by reasoning over both facts.

The following prompt is used to formulate the multi-hop questions from the paired factual triplets:

---

**Prompt Template Mhop**

Given a multi-hop fact tuple in the format (entity-0, relation-1, [MASKED-ENTITY-1], relation-2, entity-2) and a description of each relation, generate a multi-hop question-answer pair where the answer is "entity-2." Ensure that the question respects the directional nature of each relation, especially relation-2, so that it logically reflects how the masked entity-1 relates to entity-2. The question should be structured to uniquely lead to entity-2 as the answer.
Instructions for Generation:

1. Construct the Question:

   - Design the question to identify entity-2 by logically combining relation-1 and relation-2.
   - Respect the directionality of each relation as described: if relation-2 describes a relation where [MASKED-ENTITY-1] is the subject and entity-2 is the object (e.g., "is a member of"), ensure this direction is reflected in the question.
   - Avoid reversing the direction of relation-2, and confirm the natural flow between [MASKED-ENTITY-1] and entity-2.
   - Include context from entity-0 to make the question clear and specific.
   - Do not reference [MASKED-ENTITY-1] in the answer; only entity-2 should be the answer.

2. Format: Provide the output in the following format:

   - Q: [Constructed Question]
   - A: [entity-2]

Examples:
Fact Tuple: Yasis Malih, place of birth, [MASKED-ENTITY-1], member of, Organization of World Heritage Cities Relation Descriptions: Place of birth of this person., Membership of this place. Q: Which organization is the place of birth of Yasis Malih a member of? A: Organization of World Heritage Cities
Fact Tuple: Intelligent Land Investments, instance of, [MASKED-ENTITY-1], owned by, entrepreneur Relation Descriptions: Instance type of this entity., Owner of this type. Q: Who owns the instance of Intelligent Land Investments? A: Entrepreneur
Fact Tuple: Josef Novak, country of citizenship, [MASKED-ENTITY-1], highest point, Kékes Relation Descriptions: Country of citizenship of this person., Highest point of this place. Q: What is the highest point of the country of citizenship of Josef Novak? A: Kékes
Fact Tuple: Pholidaster squamatus, taxon author, [MASKED-ENTITY-1], field of work, starfish Relation Descriptions: Taxon author of this species., Field of work of this person. Q: What is the field of work of the taxon author of Pholidaster squamatus? A: Starfish
Task:
According to these instructions and examples, provide a multi-hop QA pair for the following fact tuple:
Fact Tuple: {}, {}, [MASKED-ENTITY-1], {}, {} Relation Descriptions: {}, {}

---

# B. Benchmark Description

This section provides additional insights into the generated benchmark, including its update topics, question types, temporal aspects, and specificity analysis. These analyses aim to highlight the benchmark's diversity, temporal resolution, and suitability for evaluating knowledge integration methods.

## B.1. Update Topics

To better understand the focus of the updates in each timestep, we analyze the subjects, relations, and objects of the extracted factual triplets. Figure 8 presents word clouds for each timestep, revealing varying topical focuses across the benchmark. For example, in Timestep 7 (2024/06/01–2024/06/20), terms like *solar eclipse* frequently appear, indicating a significant number of updates related to this event. This diversity underlines the richness and real-world relevance of the generated benchmark.

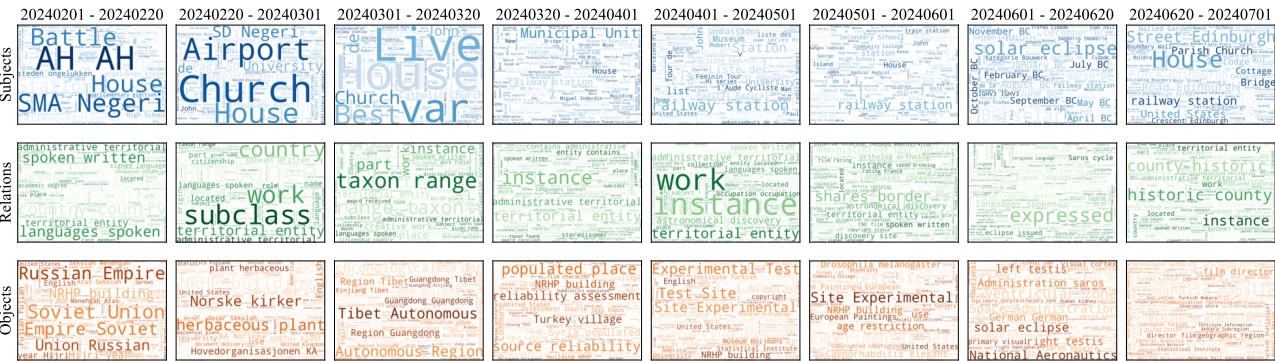

*Figure 8.* Topic word clouds for each timestep in the benchmark, illustrating the diversity of subjects, relations, and objects across different update intervals. Each row represents a component of the factual triplets (subjects, relations, and objects, respectively), while each column corresponds to a specific timestep. Notable patterns include a focus on specific events, such as "solar eclipse" in Timestep 7 (20240601–20240620), and varying distributions of topics across timesteps, emphasizing the richness and real-world relevance of the benchmark.

## B.2. Mhop Questions

To analyze the characteristics of multi-hop (mhop) questions, we evaluate the performance of LLMs on these questions without updating the models. In the top row of Figure 9, we plot the average accuracy of models on mhop questions as a function of their performance on the individual hops. The x-axis represents the accuracy on questions generated from the first part of the mhop tuple, while the y-axis shows the accuracy on the second part. The color of the markers corresponds to the overall multi-hop accuracy.

The results demonstrate a strong dependence on the accuracy of the second hop, which directly determines the final answer. These findings validate the feasibility of the mhop questions, showing that if the individual facts are known to the model, answering the multi-hop question is possible. To probe for shortcut-driven reasoning (Trivedi et al., 2020), we additionally compare multi-hop performance to performance on the individual hops (see Figure 9). While rare, we observe a small fraction of correctly answered multi-hop questions despite incorrect answers to one of the hops, indicating isolated cases of surface-level or heuristic reasoning.

## B.3. Temporal Analysis

We provide an in-depth analysis of the temporal characteristics of the benchmark. Figure 10 shows the number of samples binned by year. The right subplot zooms into updates dated after 2000 for improved visibility. While the benchmark contains updates from past years, a significant portion is dated to 2024, reflecting its focus on current and relevant knowledge.

Figure 11 presents the accuracy of LLMs binned by the timestamp of the updates. For updates dated between 1800 and 2022, performance remains stable, albeit with higher variance for older samples due to fewer data points. However, updates from 2023 and 2024 show a marked drop in accuracy, emphasizing the benchmark's relevance for studying factual knowledge integration in areas where LLMs struggle most.

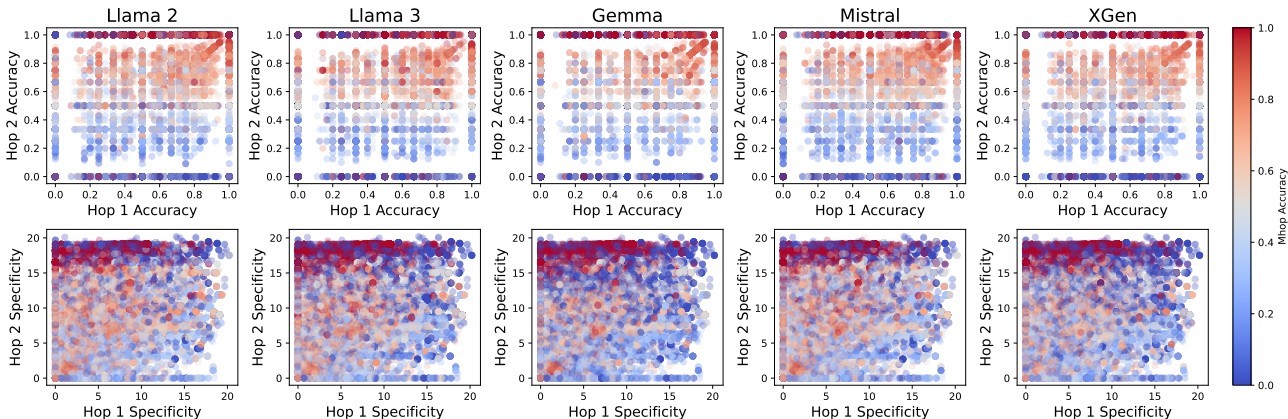

*Figure 9.* Multi-hop (mhop) question accuracy analysis for five models (Llama 2, Llama 3, Gemma, Mistral, XGen). The top row shows the relationship between the accuracy of answering questions from the first (Hop 1) and second (Hop 2) parts of mhop factual tuples and overall mhop accuracy (color-coded). Hop 2 accuracy strongly correlates with mhop question accuracy, highlighting its critical role in multi-hop reasoning. The bottom row explores the relationship between specificity (measured via entity-specificity scores) and accuracy for Hop 1 and Hop 2. Higher mhop accuracy is generally linked to lower specificity, emphasizing the challenge of highly specific entities.

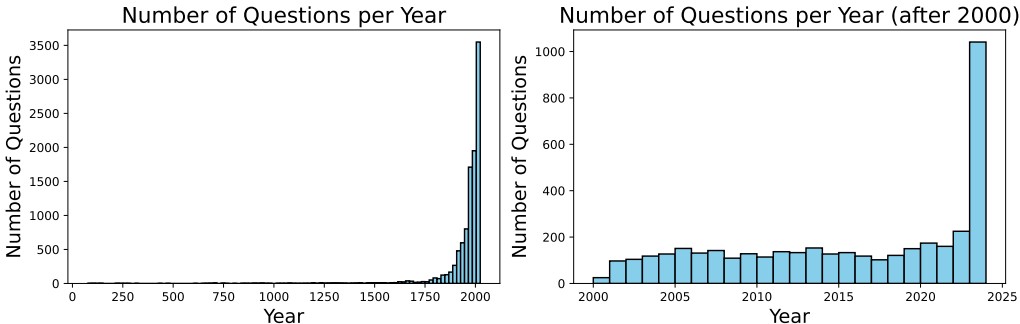

*Figure 10.* Histograms showing the distribution of the number of questions per year in the benchmark. The left plot provides an overview spanning all recorded years, highlighting a sharp increase in questions for recent years. The right plot zooms into the years after 2000, demonstrating a steady growth in questions over time, culminating in a significant spike in 2024, reflecting the focus on current updates in the benchmark.

### B.4. Specificity Analysis

The benchmark contains a mix of general and highly specific questions. For instance, general questions like "*Which continent is Tokyo a part of?*" contrast with highly specific ones like "*What is the spacing of the rails at Tire railway station?*" Quantifying question specificity is crucial, as highly specific questions may be challenging for LLMs regardless of their knowledge cutoff.

We quantify specificity using two measures:

1. **Subject and Object Counts**: Frequency of the subject and object entities in the *v4_dolma-v1_7_llama* training corpus, derived using the InfiniGram API.

2. **Subject and Object Views**: Aggregated Wikipedia page views of the subject and object entities from 01.03.2023 to 01.07.2024, following Khodja et al. (2024).

Figure 12 shows the distribution of specificity scores. The count-based measure skews towards lower frequencies (higher specificity) for subjects, while objects are more evenly distributed. By contrast, the view-based measure produces smoother distributions for both entities.

We also analyze the relationship between specificity and model performance. Figure 13 shows no clear link between Wikipedia page views and accuracy, while Figure 14 reveals a consistent trend: questions with low specificity scores

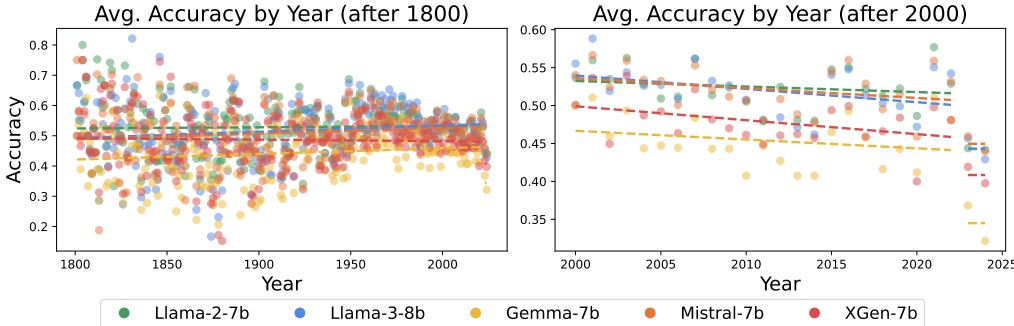

*Figure 11.* Average accuracy of five evaluated models on questions binned by their fact timestamp. The left plot includes all questions dated after 1800, showing stable performance across historical facts. The right plot focuses on questions after 2000, revealing a performance decline for facts dated to 2023 and 2024. This drop highlights the challenge of addressing more recent updates, emphasizing the relevance of the benchmark for studying factual knowledge integration in modern contexts.

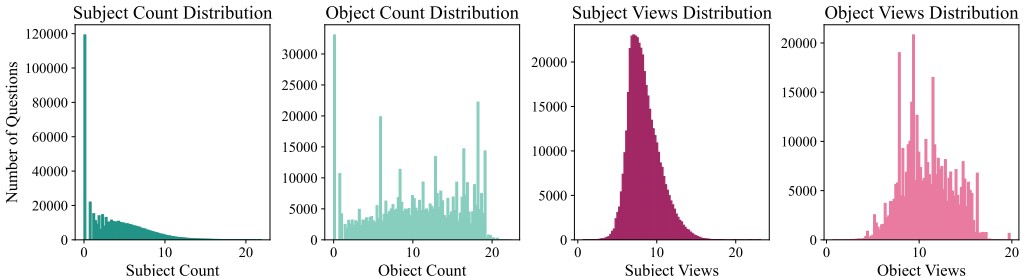

*Figure 12.* Distributions of specificity measures for subjects and objects in the benchmark. The left two plots show the frequency-based specificity, measured as the count of occurrences in a reference LLM training corpus (Infini Gram API). The distributions indicate that subjects tend to have lower frequency counts (higher specificity), while objects are more evenly distributed. The right two plots represent specificity based on Wikipedia view counts aggregated over 16 months. These distributions are smoother and centered around a similar mean for both subjects and objects, highlighting different characteristics of the two specificity measures.

(frequent entities) achieve higher accuracy than those with high specificity scores (rare entities). This suggests that specificity, as measured by training corpus frequency, is a critical factor influencing model performance.

Further analysis revealed structural biases in the view-based specificity measure. For instance, questions about languages often receive high specificity scores based on page views despite being general in nature. By contrast, the count-based measure accurately reflects their generality due to the frequent occurrence of language names in training corpora.

## C. Knowledge Integration

### C.1. Problem Formulation

The task studied in this paper is the large-scale, sequential integration of factual knowledge into a pre-trained language model, denoted as $f^0$. The model $f^0$, with frozen parameters, is trained on a dataset $D_{\text{train}}$ and deployed to process a stream of factual updates $[u_1, u_2, \ldots, u_T]$, where each update $u_t = (q_t, a_t)$ consists of a question $q_t$ and its corresponding factual answer $a_t$. These updates are grouped into $B$ sequential batches, represented as $[U_1, U_2, \ldots, U_B]$, where $U_b = \{u_{b,1}, u_{b,2}, \ldots, u_{b,N_b}\}$ is the set of updates in batch $b$, and $N_b$ is the number of updates in that batch.

At each timestep $b$, the model $f^{b-1}$, which incorporates updates from all prior batches $U_{<b} = \bigcup_{i=1}^{b-1} U_i$, is updated with the new batch $U_b$, resulting in the updated model $f^b$. The goal of this process is to enable $f^b$ to integrate the new updates, retain previously integrated knowledge, and maintain its general performance. The objectives of this sequential knowledge integration task can be summarized as follows:

**Successful Update Integration**: After processing $U_b$, the model must correctly answer all questions in $U_b$, ensuring that the newly introduced knowledge is successfully integrated:

$$f^b(q_t) = a_t, \quad \forall u_t \in U_b.$$

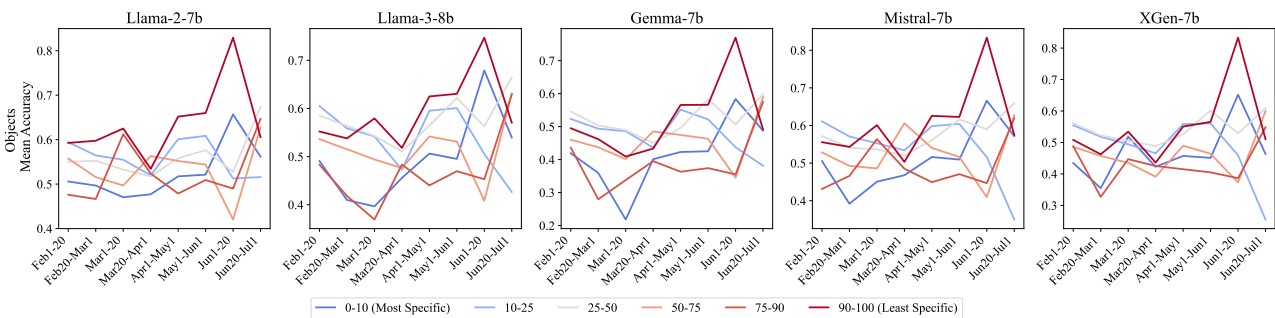

*Figure 13.* Mean accuracy of object-specific questions across specificity bins, measured using Wikipedia page views, for five evaluated models over different timesteps. The specificity bins range from the most specific (0–10, blue) to the least specific (90–100, red). While no clear trend emerges between specificity and performance, fluctuations in accuracy are observed across timesteps and specificity bins, indicating variability in how models handle questions of different specificity levels.

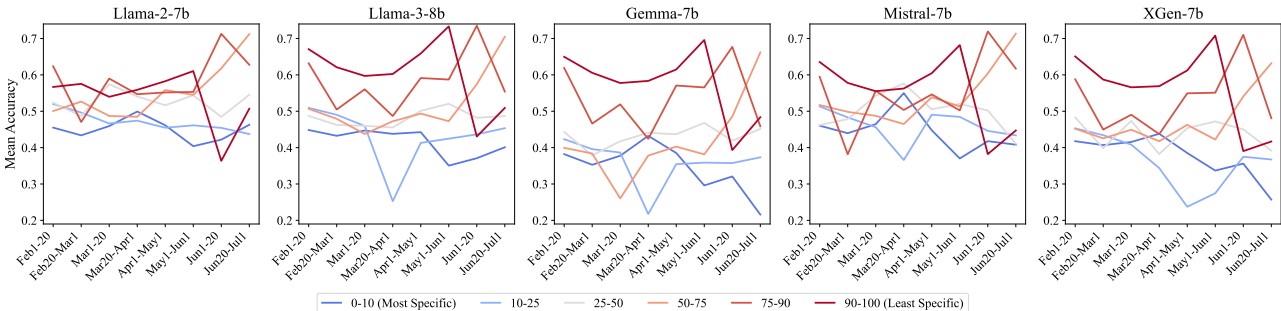

*Figure 14.* Mean accuracy of object-specific questions across specificity bins, measured using Infinigram counts, for five evaluated models over different timesteps. Specificity bins range from the most specific (0–10, blue) to the least specific (90–100, red). The results indicate a consistent trend where lower specificity (higher Infinigram counts) is associated with better model performance, with the least specific bin showing the highest accuracy across all models and timesteps. This suggests that questions involving more common entities in the training corpus are easier for the models to answer.

**Retention of Prior Updates**: The model must retain the factual updates from all previously processed batches, maintaining consistent predictions:

$$f^b(q_t) = a_t, \quad \forall u_t \in U_{<b}.$$

**Generalization of Updates**: The updated model must correctly answer rephrased questions from $D_{\text{rephrased}}$, which evaluate the generalization ability of the integrated updates:

$$f^b(q_t) = a_t, \quad \forall q_i \in D_{\text{rephrased}}.$$

**Application of Updates**: The model must utilize integrated updates for reasoning tasks, answering multi-hop evaluation questions in $D_{\text{mhop}}$ correctly:

$$f^b(q_t) = a_t, \quad \forall q_i \in D_{\text{mhop}}.$$

**Locality of Updates**: The model must preserve its predictions on unrelated facts from a control dataset $D_{\text{locality}}$, ensuring that updates do not cause spillover or degrade performance on unaffected knowledge:

$$f^b(q_i) = f^0(q_i), \quad \forall q_i \in D_{\text{locality}}.$$

To evaluate the performance of the model during this sequential update process, we monitor five key metrics after processing each batch $U_b$ : (1) the accuracy on the current batch $U_b$, (2) the retention of accuracy on prior updates from $U_{<b}$, (3) the accuracy on the rephrased questions $D_{\text{locality}}$, (4) the accuracy on multi-hop questions $D_{\text{mhop}}$, and (5) the locality of predictions on unrelated facts in $D_{\text{locality}}$. Additionally, forgetting is monitored by continuously evaluating the model on previously updated batches as new batches are integrated. This sequential process ensures that knowledge updates accumulate over time, creating a challenging setting for lifelong factual knowledge integration.

## C.2. Detailed Results

| Approach | Model | Update | Rephrase | Personas | Mhop | Locality |
|---|---|---|---|---|---|---|
| | llama-2-7b | 49.90 (±2.30) | 47.74 (±3.65) | 41.18 (±2.26) | 43.91 (±6.02) | 48.62 (±2.41) |
| | llama-3-8b | 46.87 (±3.30) | 44.32 (±4.09) | 40.12 (±2.85) | 44.29 (±8.37) | 46.92 (±2.54) |
| Pre-Update | mistral-7b | 48.83 (±2.26) | 47.02 (±3.32) | 40.48 (±2.24) | 42.19 (±6.18) | 47.97 (±2.75) |
| | gemma-7b | 40.61 (±3.33) | 38.66 (±4.14) | 33.08 (±3.10) | 35.40 (±7.36) | 40.86 (±2.92) |
| | xgen-7b | 40.80 (±1.83) | 38.64 (±3.33) | 33.61 (±2.45) | 34.87 (±5.58) | 40.49 (±2.13) |
| | llama-2-7b | 96.17 (±0.60) | 84.92 (±5.76) | 72.62 (±2.43) | 51.08 (±6.66) | 56.29 (±5.09) |
| | llama-3-8b | 99.18 (±0.16) | 86.19 (±6.37) | 74.55 (±2.71) | 49.67 (±8.66) | 56.26 (±5.45) |
| RAG | mistral-7b | 99.25 (±0.19) | 85.89 (±6.07) | 74.83 (±2.50) | 51.88 (±6.71) | 56.97 (±5.38) |
| | gemma-7b | 95.06 (±0.73) | 81.98 (±6.15) | 69.34 (±2.86) | 44.84 (±8.02) | 49.33 (±5.07) |
| | xgen-7b | 95.98 (±0.46) | 82.75 (±6.11) | 71.45 (±2.94) | 45.54 (±6.82) | 51.41 (±5.22) |
| | llama-2-7b | 63.07 (±8.10) | 56.54 (±7.03) | 47.19 (±7.05) | 43.72 (±10.17) | 46.10 (±3.42) |
| | llama-3-8b | 62.84 (±4.54) | 56.27 (±5.43) | 46.74 (±5.27) | 43.24 (±9.06) | 45.96 (±2.88) |
| LoRA-FT | mistral-7b | 29.21 (±22.71) | 25.65 (±20.85) | 21.89 (±17.57) | 20.29 (±18.72) | 21.43 (±16.47) |
| | gemma-7b | 57.38 (±5.86) | 51.33 (±6.10) | 42.00 (±5.07) | 38.96 (±10.06) | 38.35 (±2.33) |
| | xgen-7b | 51.53 (±8.10) | 46.21 (±8.17) | 38.07 (±8.76) | 37.38 (±10.20) | 36.48 (±2.85) |
| | llama-2-7b | 67.36 (±3.12) | 62.01 (±4.71) | 56.64 (±2.93) | 50.29 (±7.17) | 58.40 (±4.44) |
| | llama-3-8b | 64.84 (±2.29) | 59.79 (±4.56) | 55.65 (±2.24) | 47.68 (±8.10) | 58.24 (±4.26) |
| LoRA-Merge | mistral-7b | 61.84 (±1.46) | 57.17 (±3.82) | 52.60 (±1.66) | 49.24 (±6.92) | 54.68 (±3.20) |
| | gemma-7b | 60.13 (±4.57) | 54.87 (±6.29) | 49.84 (±3.70) | 44.64 (±8.09) | 53.26 (±5.80) |
| | xgen-7b | 62.38 (±2.50) | 57.57 (±4.49) | 51.03 (±2.82) | 45.16 (±7.51) | 53.72 (±5.01) |

*Table 5.* Detailed evaluation results of different approaches (Pre-Update, RAG, LoRA-FT, and LoRA-Merge) across five metrics: Update, Rephrase, Personas, Mhop, and Locality. Results are presented for five models (Llama-2-7b, Llama-3-8b, Mistral-7b, Gemma-7b, and XGen-7b) and are averaged across all timesteps, weighted by their respective sizes. Each value represents mean accuracy with standard deviation (±). This table highlights the performance variability of different approaches and models across metrics.

Table 5 provides a comprehensive overview of the performance of the evaluated approaches - Pre-Update, RAG, LoRA-FT, and LoRA-Merge - across the five key metrics: Update, Rephrase, Personas, Mhop, and Locality. The results are reported for five different models (Llama-2-7b, Llama-3-8b, Mistral-7b, Gemma-7b, and XGen-7b) and averaged across all timesteps, weighted by their respective sizes. Each entry in the table displays the mean accuracy with standard deviation over the eight timesteps, offering detailed insights into the effectiveness and stability of each approach under the lifelong knowledge integration setting.

### C.3. Retrieval Augmented Generation (RAG)

The RAG baseline augments a pre-trained language model $f^0$ with a non-parametric memory $M$, enabling retrieval-augmented generation. The memory $M$ stores factual updates and their embeddings, allowing the model to access relevant information dynamically. The components and process of the RAG baseline are described below:

**Memory Construction.** For each batch of updates $U_b = \{u_{b,1}, u_{b,2}, \ldots, u_{b,N_b}\}$, where $u_{b,i} = (q_{b,i}, a_{b,i})$, the memory is constructed as follows:

- Initialization: Each update question $q_{b,i}$ and its corresponding answer $a_{b,i}$ are stored in memory.

- Embedding Computation: The embedding model $g$ encodes each question into a fixed-dimensional vector: $e_{b,i} = g(q_{b,i})$.

- Memory Entry: Each memory entry is stored as a tuple $(q_{b,i}, a_{b,i}, e_{b,i})$, added to the memory $M : M \leftarrow M \cup$

$\{(q_{b,i}, a_{b,i}, e_{b,i})\}.$

**Test-Time Retrieval.** At test time, given an input query $q_{\text{test}}$, the RAG baseline retrieves relevant entries from $M$ and augments the input context for the language model:

- Embedding Computation for Query: The query embedding is computed as: $e_{\text{test}} = g(q_{\text{test}})$.

- Nearest Neighbor Retrieval: The top-k most similar entries are retrieved based on cosine similarity between $e_{\text{test}}$ and stored embeddings $e_{b,i}$:

$$\text{Retrieve } \{(q_{b,j}, a_{b,j})\}j = 1^k \text{ such that } \text{sim}(e_{\text{test}}, e_{b,j}) \text{ is maximized.}$$

Efficient retrieval is implemented using the Annoy solver (Bernhardsson, 2018) for hierarchical navigable small world (HNSW) graphs.

- Context Construction: The retrieved questions $\{q_{b,j}\}_{j=1}^{k}$ and answers $\{a_{b,j}\}_{j=1}^{k}$ are prepended to $q_{\text{test}}$ to form the augmented context:

$$C_{\text{test}} = \text{Concat}([q_{b,1}, a_{b,1}, \ldots, q_{b,k}, a_{b,k}, q_{\text{test}}]).$$

**Response Generation.** The augmented context $C_{\text{test}}$ is passed to the language model $f$, which generates the response $a_{\text{test}} : a_{\text{test}} = f(C_{\text{test}})$.

This retrieval-augmented approach provides a simple yet scalable mechanism for integrating and accessing factual updates during inference, making it well-suited for the lifelong knowledge integration task.

**Parameter Selection and Ablation.** For the main experiments, $k = 2$ was chosen to enable effective multi-hop reasoning while keeping the context length manageable. Figure 15 shows an ablation study over $k$, using subsets of 5,000 updates per timestep with $k$ ranging from 1 to 100. The results indicate that increasing $k$ improves performance across metrics, especially for rephrased questions, as it increases the likelihood of retrieving relevant entries. The performance on the update sets remains stable regardless of $k$, as they directly match the stored memory entries. Larger $k$ values result in higher computational overhead, with $k = 100$ increasing the forward pass time by 3x compared to $k = 1$. Balancing performance gains and computational costs is thus critical.

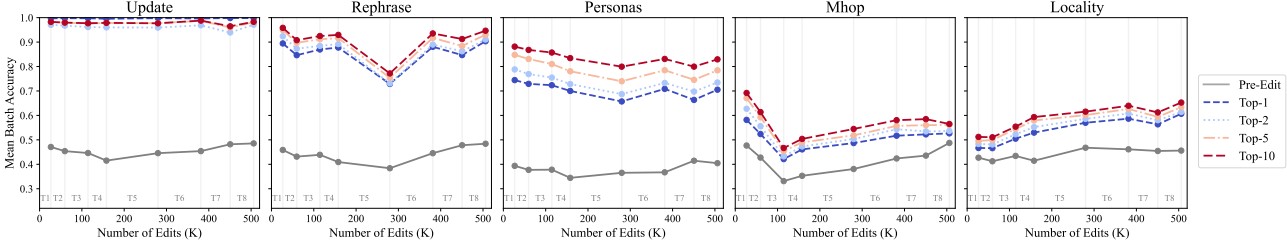

*Figure 15.* Ablation results for the top-k parameter of the RAG baseline, evaluating update, rephrase, personas, mhop, and locality sets across 500k updates. Higher top-k values improve performance on rephrase, personas, and mhop sets due to increased retrieval coverage, while update set accuracy remains consistent across top-k values. The locality set shows marginal gains with higher top-k values, indicating reduced spillover effects. However, increasing top-k also leads to greater computational overhead, highlighting the trade-off between retrieval depth and efficiency.

**Computational Overhead.** We evaluate the test-time compute of RAG compared to pre-update models and LoRA fine-tuning. Corrected by batch size, forward pass times serve as the measure of computational overhead. Figure 16 shows that LoRA finetuning introduces negligible overhead, with minimal variations attributed to outliers. Meanwhile, RAG approximately doubles the forward pass time across all models due to the additional retrieval and larger context. This tradeoff highlights RAG's strength in dynamically integrating factual updates but emphasizes the need to balance retrieval efficiency and test-time compute for large-scale applications.

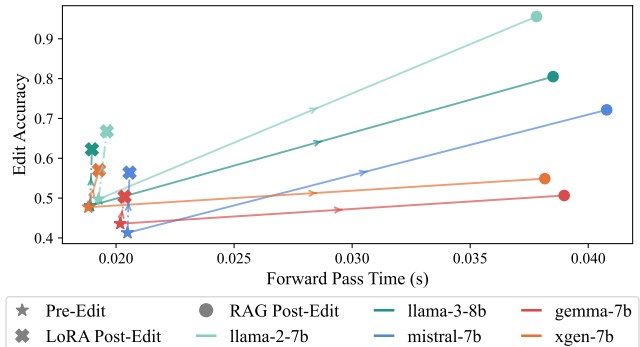

*Figure 16.* Trade-off between forward pass time (x-axis) and edit accuracy (y-axis) for RAG and LoRA across five models. Stars denote pre-edit performance, while post-edit performance for RAG and LoRA is represented by circles and crosses, respectively. RAG achieves higher edit accuracy at the cost of increased forward pass time, nearly doubling the average inference latency compared to LoRA. LoRA introduces minimal additional computational overhead while maintaining moderate accuracy improvements over the pre-edit baseline, highlighting its efficiency in resource-constrained settings.

### C.4. Continual Finetuning

We evaluate the task of continual finetuning using LoRA (Hu et al., 2021) under two distinct configurations: LoRA-FT, where adapters are continuously trained across sequential updates, and LoRA-Merge, which incorporates weight merging and interpolation to balance newly integrated knowledge with prior retained knowledge. Both configurations aim to integrate large-scale factual updates efficiently while minimizing catastrophic forgetting and ensuring stable performance over time.

**Continual LoRA Fine-Tuning (LoRA-FT)**

1. Initialization: For the initial model $f^0$, LoRA adapters $\Delta^0$ with learnable parameters are initialized and attached to specific layers in the model. The updated model $f^1$ is defined as: $f^1(x) = f^0(x) + \Delta^1(x)$, where $\Delta^1(x)$ represents the output of the LoRA adapter.

2. Sequential Update Process: For each batch of updates $U_b$ , the LoRA adapters are trained on the corresponding question-answer pairs $U_b = \{(q_{b,i}, a_{b,i})\}_{i=1}^{N_b}$ . The LoRA parameters $\Delta^b$ from the previous timestep are reused and further optimized: $\Delta^b \leftarrow \Delta^{b-1} - \eta\nabla_{\Delta^{b-1}}\mathcal{L}(f^{b-1}(q_{b,i}), a_{b,i})$, where $\mathcal{L}$ is the loss function, and $\eta$ is the learning rate. Cosine learning rate scheduling with warmup is employed during training, with a fixed number of 10 epochs per batch.

3. Model Behavior: The updated model at timestep $b$ is defined as: $f^b(x) = f^0(x) + \Delta^b(x)$, where $\Delta^b$ accumulates adaptations from all previous timesteps $U_{<b}$.

**LoRA Fine-Tuning with Merging (LoRA-Merge)**

1. Initialization: Similar to LoRA-FT, LoRA adapters $\Delta^0$ are initialized and attached to the model $f^0$.

2. Weight Merging and Interpolation: After training the LoRA adapters $\Delta^b$ on the batch $U_b$, the adapted model weights $w_{\text{adapted}}$ are merged into the base model weights $w_{\text{base}}$, creating an interpolated model: $w_{\text{merged}} = \alpha w_{\text{base}} + (1-\alpha)w_{\text{adapted}}$, where $\alpha$ is the interpolation factor.

3. Sequential Update Process: The merged weights $w_{\text{merged}}$ serve as the starting point for training LoRA adapters on the next batch $U_{b+1}$.

4. Model Behavior: This process ensures that each timestep balances newly integrated updates with previously retained knowledge, leveraging the interpolation factor $\alpha$ to control the degree of retention.

**Ablations and Results**

**Parameter Selection LoRA-FT.** We conduct an ablation on the rank of the LoRA matrices in the LoRA-FT setting, comparing a high-rank configuration (rank = 64) with a low-rank configuration (rank = 4). As shown in Figure 17, the

low-rank setting consistently outperforms the high-rank setting across all metrics. Additionally, the high-rank configuration exhibits greater instability, with instances of catastrophic failure, such as breaking the Mistral model. The results also reveal increased forgetting in the high-rank setting compared to the low-rank setting.

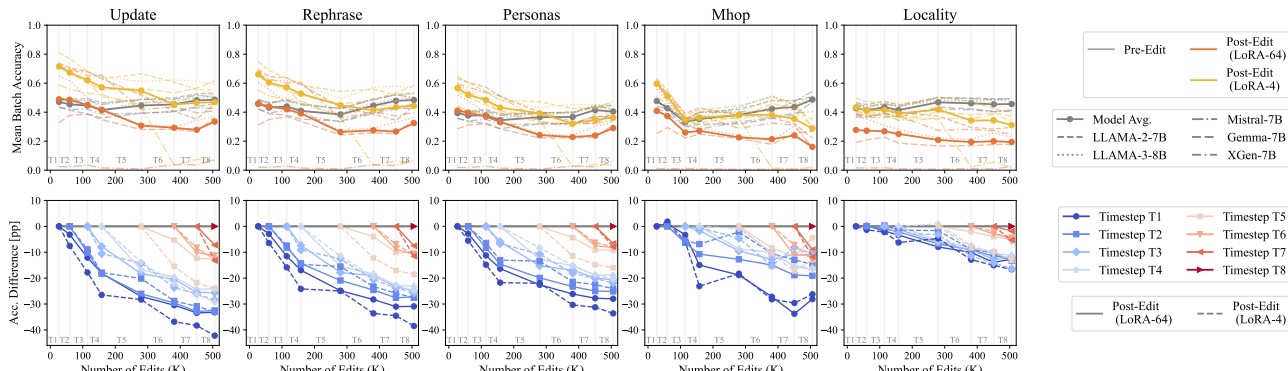

*Figure 17.* Performance of continual LoRA fine-tuning with high-rank (LoRA-64) and low-rank (LoRA-4) configurations across evaluation sets. The top row shows the mean batch accuracy for update, rephrase, personas, mhop, and locality sets, while the bottom row displays accuracy differences for individual timesteps. Low-rank configurations maintain competitive performance initially but degrade over time, while high-rank configurations show instability, including catastrophic failures in certain models.

**Parameter Selection LoRA-Merge.** We further investigate the impact of the interpolation factor $\alpha$ in the LoRA-Merge setting, as shown in Figure 18. We evaluate two configurations: one where the base model and adapted model are equally weighted ($\alpha = 0.5$) and another where the base model is given higher weight ($\alpha = 0.75$). The results indicate that higher weight given to the adapted model ($\alpha = 0.5$) leads to stronger initial adaptation but results in performance decay over time. Conversely, prioritizing the base model ($\alpha = 0.75$) provides more stable performance across timesteps.

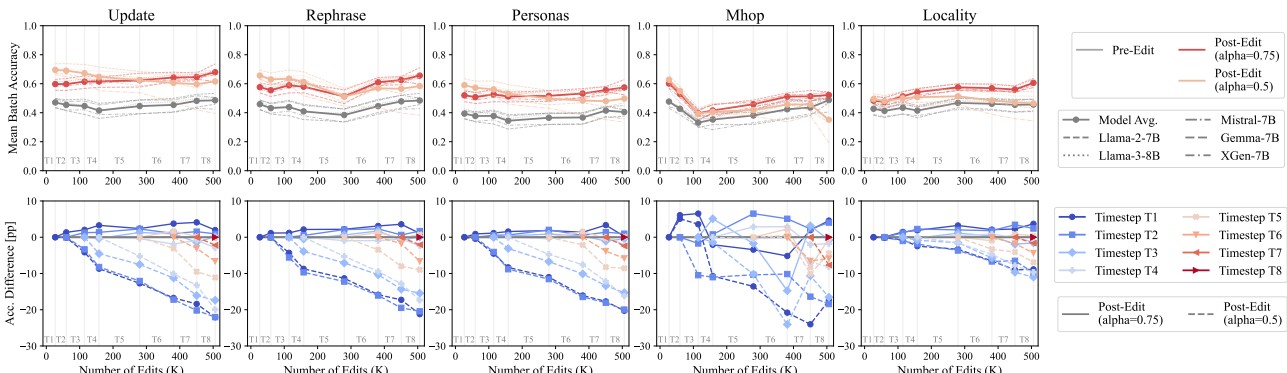

*Figure 18.* Ablation study on the interpolation factor $\alpha$ in the LoRA-Merge setting, comparing $\alpha = 0.75$ (red) and $\alpha = 0.5$ (orange). The top row shows the mean batch accuracy for the update, rephrase, personas, mhop, and locality sets, while the bottom row depicts accuracy differences (in percentage points) across timesteps. Higher weight on the base model ($\alpha = 0.75$) provides more stable performance and mitigates forgetting, while equal weighting ($\alpha = 0.5$) achieves higher initial performance but leads to greater degradation, particularly on the mhop and locality sets. Bold lines represent average model performance, with lighter lines for individual models.

### C.5. Knowledge Editing

The knowledge editing task involves updating a pre-trained language model $f^0$, with frozen parameters trained on $D_{\text{train}}$, to modify its behavior for specific inputs $q_t$ to produce updated outputs $a_t$. These updates must be incorporated without retraining the entire model and while preserving the model's performance on unrelated tasks and previously integrated updates. The key requirements of the task are:

1. Successful Knowledge Integration: The edited model $f^b$ should correctly predict $a_t$ for all edited questions $q_t$.

2. Retention of Unrelated Knowledge: The edits should not degrade the model's performance on unrelated queries $q_i \in D_{\text{locality}}$, ensuring $f^b(q_i) = f^0(q_i)$.

3. Consistency Across Prior Edits: The model should maintain the correctness of previous edits $q_{<t}$, ensuring $f^b(q_{t\prime}) = a_{t\prime}$ for all $t\prime < t$.

The following describes three prominent knowledge editing methods—ROME, MEMIT, and WISE—using this notation.

**Local Modification Approaches.** Local modification approaches focus on directly altering the parameters of the model at specific layers to integrate new knowledge while minimizing unintended side effects. This category includes ROME and MEMIT.

**Rank-One Model Editing (ROME).** ROME (Meng et al., 2023) is a method for performing rank-one updates to a specific layer in the model. It focuses on precisely editing the causal pathways responsible for a particular factual association.

- Objective: Modify the model $f^b$ such that, for an update $u_t = (q_t, a_t)$, the edited model satisfies: $f^b(q_t) = a_t$, while minimizing interference with unrelated predictions.

- Layer Selection: Identify the most causally relevant layer $l^*$ for the given update by analyzing the model's causal mediation.

- Update Rule: Modify the weights $W_{l^*}$ in the key-value projection as: $W_{l^*} \leftarrow W_{l^*} + \Delta W$, where $\Delta W = uv^\top$ is a rank-one update computed from the desired output $a_t$ and intermediate activations at layer $l^*$.

**Multistep Editing for Memory-Intensive Tasks (MEMIT).** MEMIT (Meng et al., 2022) extends ROME to handle multiple updates simultaneously and distributes them across multiple layers.

- Objective: Modify the model $f^b$ to incorporate a batch of updates $U_b = \{(q_{b,i}, a_{b,i})\}_{i=1}^{N_b}$, such that: $f^b(q_{b,i}) = a_{b,i}, \quad \forall u_{b,i} \in U_b$, while maintaining performance on unrelated queries.

- Layer Selection: Select a set of layers $\{l_1, l_2, \ldots, l_k\}$ for distributing the updates to reduce interference.

- Update Rule: For each selected layer $l$, apply a weight update: $W_l \leftarrow W_l + \Delta W_l$, where $\Delta W_l$ is computed via low-rank decomposition to ensure computational efficiency and scalability.

Local modification approaches are effective for small-scale or isolated updates but face challenges with scalability and catastrophic forgetting in sequential update scenarios.

**Lifelong Knowledge Editing Approaches.** Lifelong knowledge editing approaches are designed to handle sequential, large-scale updates while mitigating catastrophic forgetting and maintaining stability over time. This category includes WISE.

**Weight Injection for Structured Editing (WISE).** WISE (Wang et al., 2024a) leverages a semi-parametric memory to store layer-specific weight adjustments for each update, enabling reversible and modular edits.

- Objective: For a sequence of updates $[u_1, u_2, \ldots, u_T]$, modify the model $f^b$ such that $f^b(q_t) = a_t, \quad \forall t \leq T$, while ensuring $f^b(q_i) = f^0(q_i), \quad \forall q_i \in D_{\text{locality}}$, and retaining accuracy on prior updates $u_{<t}$.

- Memory Construction: For each update $u_t = (q_t, a_t)$, compute weight adjustments $\Delta W_l$ for specific layers $l$. Store these adjustments in a memory: $M_l \leftarrow M_l \cup \Delta W_l, \quad \forall l \in L$, where $L$ is the set of edited layers.

- Inference with Memory: During inference, apply the stored adjustments $\Delta W_l$ to the forward pass: $f^b(x) = f^0(x) + \sum_{l \in L} \Delta W_l(x)$.

- Scalability: By maintaining a modular memory, WISE avoids catastrophic forgetting and enables scalability to large sequences of updates.

Lifelong editing approaches like WISE are better suited for large-scale, sequential updates, though they may still face challenges with long-term consistency and retrieval efficiency in memory-based systems.

**Knowledge Editing for Lifelong Knowledge Integration** Figure 19 provides additional results on the application of knowledge editing techniques to the lifelong knowledge integration task. The figure presents the accuracy on the update set for the first 500 updates and the entire first timestep (26K updates) for the Llama-2 model. The results highlight the rapid performance deterioration of local modification approaches, with accuracy dropping significantly within the first 250 updates. While WISE performs comparably to RAG for fewer than 500 updates, its performance deteriorates over the first 10K updates, eventually converging to pre-update performance levels. This behavior underscores the challenges faced by knowledge editing approaches in handling larger-scale, sequential updates.

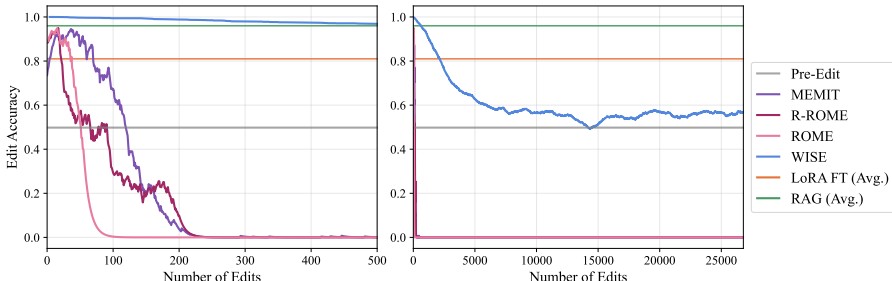

*Figure 19.* Update accuracy of knowledge editing approaches (MEMIT, R-ROME, ROME, and WISE) compared to LoRA-FT and RAG baselines across the first 500 updates (left) and the full first timestep (26k updates, right). Local modification methods (MEMIT, R-ROME, ROME) rapidly degrade within the first 250 updates, converging to near-zero performance. WISE initially performs on par with RAG for fewer than 500 updates but declines over the first 10k updates, converging to pre-update accuracy, highlighting its limitations in maintaining update accuracy for larger-scale knowledge integration.

In the main experiments, MEMIT exhibits model collapse after approximately 250 sequential edits. This contrasts with the original MEMIT paper (Meng et al., 2022), which demonstrates successful integration of up to 10K edits. To investigate this discrepancy, we conduct an ablation study where we vary the batch size (i.e., the number of edits processed simultaneously) for MEMIT on Llama-2 using the first 10K updates of the `WikiBigEdit` benchmark. The results, presented in Figure 20, reveal that MEMIT avoids model collapse when edits are processed in larger batches, with the performance remaining stable when all 10K edits are applied at once. However, the overall update accuracy in this setting remains below the pre-edit performance of the model, highlighting a trade-off between stability and efficacy.

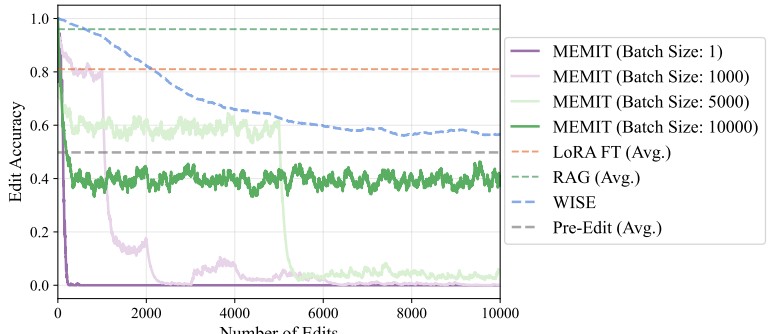

*Figure 20.* Impact of batch size on MEMIT's performance for the first 10k updates of the `WikiBigEdit` benchmark using Llama-2. Larger batch sizes (e.g., 10k edits) prevent model collapse but result in lower update accuracy compared to the model's pre-edit performance. Smaller batch sizes (e.g., 1 or 1k edits) perform well initially but exhibit rapid degradation in subsequent updates, highlighting MEMIT's limitations for sequential, large-scale lifelong knowledge editing.

Interestingly, when applying smaller batch sizes (e.g., 1, 1K, or 5K edits per batch), MEMIT performs well for the initial batch but collapses in subsequent updates. This behavior indicates a severe limitation: while batch processing can temporarily mitigate collapse, it is impractical for real-world scenarios where edits arrive sequentially over time or at a large scale. The need to process edits as they become available underscores MEMIT's unsuitability for sequential, large-scale lifelong knowledge editing. These findings emphasize the importance of developing methods that remain stable and effective under practical constraints, where timely incorporation of updates is crucial.

In Figure 21, we present additional experiments with WISE conducted on three language models: Llama-2, Llama-3, and Mistral. These experiments focus on the first timestep of the WikiBigEdit benchmark. The results consistently show that WISE converges to pre-update performance across all evaluated metrics within the first timestep. Although WISE avoids catastrophic forgetting, its inability to maintain the accuracy of newly integrated updates over time highlights its limitations for lifelong knowledge integration tasks.

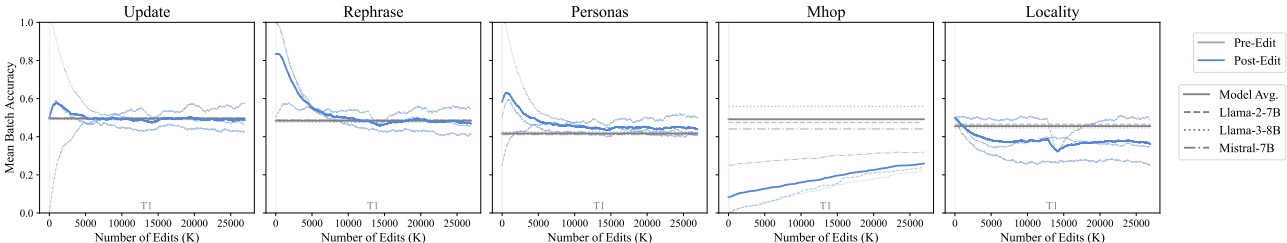

*Figure 21.* Results of WISE on three language models (Llama-2-7B, Llama-3-8B, and Mistral-7B) for the first timestep of the WikiBigEdit benchmark. Each subplot shows mean batch accuracy for the update, rephrase, personas, mhop, and locality sets over the number of updates. Post-update performance (blue) converges to pre-update levels (gray) across all metrics, highlighting WISE's limitations in sustaining accuracy after large-scale updates.

