# OpenReview forum: "WikiBigEdit: Understanding the Limits of Lifelong Knowledge Editing in LLMs"
_ICML.cc/2025/Conference — ICML 2025 poster_

### Official Review · Reviewer_UH8c · 2025-03-11

**Overall Recommendation:** 4

**Summary:**

This paper proposes a benchmark called WikiBigEdit for evaluating knowledge editing. The benchmark is constructed from Wikidata edits and includes 500k question-answer pairs. They evaluate a number of existing methods for knowledge editing and find that current techniques have limitations where continual fine-tuning and retrieval augmented methods perform better. They suggest one of their contributions is to have a fully automated extraction pipeline which continuously extracts suitable factual edits from Wikidata. The resulting WikiBigEdit spans eight time-intervals over five months (February - July 2024).

Section 3 discusses the design of WikiBigEdit. Their construction is divided into 7 steps:
1. Periodic Snapshot Acquisition ($S_{changed}, S_{unchanged}$): downloading subject-relation-object relations from Wikidata for most recent snapshots. Triplets are divided into changed and unchanged sets.
2. Initial Filtering: this step excludes triplets based on simple rules such as length of the subject or object.
3. Generation of Locality Probes ($S_{locality}$): finds pairs of triplets in changed and unchanged sets where the subject-relation is similar but the object is distinct.
4. Inclusion of multi-hop reasoning tuples ($S_{mhop}$): extracts pairs of factual triplets that are linked to each other by a shared subject/object. These pairs can be used to create multi-hop questions.
5. Generation of QA edit pairs: for the sets changed and locality GPT-3.5 and for the set of mhop facts GPT-4o-mini are utilized to create question-answer pairs. For mhop, the questions omit the middle shared entity.
6. Generation of personalized and rephrased question-answer edit pairs: to further evaluate the reasoning generalization, GPT-4o-mini is used to create question-answer variants by either rephrasing or mimicking a persona that rewrites the QA in various styles.
7. Final filtering stage: filter QAs to make sure the question contains the subject/relation but not the answer/object and answers include the object.

The paper provides various analyses of the benchmark. For example, Figure 2.a shows that the generation QAs focus on recent events although questions about older events also exist as old as the 1900s.  Figure 2.b shows that the performance of five LLMs drops after their cut-off date on the proposed benchmark.

Section 4 evaluates the performance of various knowledge editing methods. Figure 3 summarizes the performance of these methods on subsets of the evaluation. It shows while RAG outperforms all methods, continual fine-tuning methods based on LoRA perform better than other knowledge editing methods. The result about RAGs is expected as retrieving the correct answer from Wikipedia/Wikidata should be easy for RAG methods while multi-hop reasoning does not see significant improvement. Figure 4 also shows that performance of knowledge editing methods decays as the number of edits increase which is a limitation of such methods over long horizons.

## Update after rebuttal
I thank authors for their response. I encourage authors to incorporate their response into a revision, particularly the result of multi-hop evaluations and connection DiRe condition as well relation to other methods and benchmarks. I maintain a positive rating.

**Claims And Evidence:**

The paper introduces a new benchmark to evaluate knowledge editing capabilities. The paper shows the importance of the problem, discusses the construction with details, and evaluates existing methods and provides interesting insights.

**Essential References Not Discussed:**

N/A

**Experimental Designs Or Analyses:**

The design of the evaluations and comparison of models is well designed. The authors may consider testing their evaluation against the DiRe condition. See paper below:
- Trivedi, Harsh, et al. "Is multihop QA in DiRe condition? Measuring and reducing disconnected reasoning." In Proceedings of the 2020 Conference on Empirical Methods in Natural Language Processing (EMNLP)

**Methods And Evaluation Criteria:**

The selection of methods covers a wide spectrum of methods. The authors may consider including evaluation of more recent RAG methods that potentially utilize an LLM for reasoning on facts that may improve on multi-hop reasoning evaluations proposed in the paper.
- Gutiérrez, Bernal Jiménez, et al. "Hipporag: Neurobiologically inspired long-term memory for large language models." The Thirty-eighth Annual Conference on Neural Information Processing Systems. 2024.

**Other Comments Or Suggestions:**

Authors may consider discussing the relation to recent concurrent work TiC-LM and particularly TiC-Wiki where a related automatically generated continual benchmark based on Wikipedia and Wikidata over a span of 10 years has been proposed.
- Li, Jeffrey, et al. "Tic-lm: A multi-year benchmark for continual pretraining of language models."

**Other Strengths And Weaknesses:**

Strengths
- The paper is well-written, the benchmark is well-designed and provides insightful findings.

Weaknesses
- Please see above for potential comparisons and related works. Specifically, performing a variant of DiRe test may be helpful in understanding whether the multi-hop reasoning questions could be answered with a shortcut. It may be hard to apply the test directly as the QAs are written by GPT models. One potential way is to manually remove the mention of the subject from the question and evaluate if the question would be still answerable without it. Some questions about general knowledge may have this property.

**Questions For Authors:**

N/A

**Relation To Broader Scientific Literature:**

The question of how to keep the knowledge of LLMs up to date is an important research direction that requires stronger benchmarks. This work is a helpful evaluation for better understanding the limitations of LLMs and designing improved methods to keep their knowledge up to date.

**Theoretical Claims:**

No theoretical claims made.

---

> ### Author Rebuttal · Authors · 2025-04-01
>
> We thank the reviewer for their detailed and thoughtful feedback. We’re pleased they found the paper well-written, the benchmark well-designed, and the findings insightful. We appreciate the recognition of *WikiBigEdit* as a valuable contribution for evaluating real-world limitations of knowledge editing, as well as the clarity of our benchmark construction. We’re also grateful for the positive assessment of our method coverage and the relevance of our experimental insights, particularly regarding the challenges current editing techniques face over long update horizons. We address the reviewer’s suggestions and questions below.
>
> **Advanced RAG Approaches**
>
> We thank the reviewer for the helpful suggestion of including more advanced RAG approaches and for highlighting HippoRAG [1]. While HippoRAG introduces a compelling long-term memory mechanism via structured memory graphs, it assumes access to full-text documents—unlike our setting, which focuses on factual QA pairs derived from structured triplets.
>
> We agree that advanced retrieval-augmented approaches, including reasoning-augmented RAG or RAG agents, could enhance multi-hop reasoning. However, our goal is to evaluate the limits of lifelong knowledge editing in a controlled setting. We, therefore, use a simple, transparent RAG baseline to compare against editing and continual finetuning. Notably, even this baseline outperforms editing methods across most axes, though at higher inference cost.
>
> RAG’s difficulty with multi-hop reasoning is orthogonal to the challenge of integrating new knowledge over time — the focus of our work. More advanced RAG variants may improve multi-hop QA but likely require even more test-time compute.
> We will add this discussion, along with a reference to HippoRAG, to Section 4.3 of the paper.
>
> **Shortcuts in Mhop Evaluation**
>
> We appreciate the reviewer’s suggestion to consider the DiRe condition [2]. While we do not explicitly evaluate it in its original form, we conduct a related analysis in the Supplementary Material (Figure 9, top row). Specifically, we compare model performance on individual first-hop and second-hop questions to its performance on the combined multi-hop question. This serves to test whether the model relies on shortcut reasoning, i.e., answering the multi-hop question without resolving its components, which is central to the DiRe condition.
>
> Our results show that multi-hop accuracy depends heavily on correctly answering both hops. When a model fails on either, it rarely succeeds on the full question. However, a small fraction of questions (avg. 1.75% across models) are answered correctly despite failure on the first hop. These cases often involve surface-level cues, e.g.:
> * “What is the language of work or name associated with the spouse of Johann Caspar Richter?” → German
> * “In which country is the organization that Yamato Auto Works is a member of located?” → Japan
>
> These align with shortcut-driven behavior as discussed in the DiRe framework. We will clarify this connection in Section 3.1 of the revision.
>
> **Reference of Concurrent Work TiC-Wiki**
>
> We thank the reviewer for pointing us to the concurrent work TiC-Wiki [3], which shares similarities in spirit. We will include a discussion of it in the related works section. While TiC-Wiki is a valuable benchmark for continual pretraining, it covers Wikipedia/Wikidata revisions from 2008–2022 — a period largely included in the pretraining data of modern LLMs. This makes it less suitable for evaluating updates beyond the knowledge cutoff, which is the central focus of our work. In contrast, WikiBigEdit is constructed from post-cutoff updates, allowing direct evaluation of newly emerging knowledge. We view the two efforts as complementary and appreciate the chance to clarify this distinction.
>
> [1] Gutiérrez, B. J., et al. HippoRAG: Neurobiologically Inspired Long-Term Memory for Large Language Models, 2024.
>
> [2] Trivedi, Harsh, et al. Is multihop QA in DiRe condition? Measuring and Reducing Disconnected Reasoning. 2020.
>
> [3] Li, Jeffrey, et al. TiC-LM: A Multi-Year Benchmark for Continual Pretraining of Language Models, 2024.

---

> > ### Comment · Reviewer_UH8c · 2025-04-03
> >
> > I thank authors for their response. I encourage authors to incorporate their response into a revision, particularly the result of multi-hop evaluations and connection DiRe condition as well relation to other methods and benchmarks. I maintain a positive rating.

---

### Official Review · Reviewer_j8NE · 2025-03-13

**Overall Recommendation:** 3

**Summary:**

Enabling LLMs to retain up-to-date knowledge is of significant practical importance. To avoid costly full-parameter retraining, recent research has proposed various lifelong knowledge editing methods to inject new knowledge into models at minimal expense. However, these knowledge editing approaches have two notable limitations: 1. The scale of edited knowledge is relatively small; 2. The complexity of the knowledge is limited. These constraints make existing knowledge editing methods difficult to apply directly in real-world scenarios.
To better simulate real-world conditions and investigate the limitations of knowledge editing techniques, the authors designed an automated data extraction pipeline to acquire the latest world knowledge from Wikipedia. In this pipeline, they employed LLMs to rephrase the data, generating five distinct datasets collectively named WikiBigEdit. These five datasets (four of which serve as test sets) evaluate model knowledge editing effectiveness from four perspectives: 1. Ability to answer rephrased questions; 2. Ability to answer questions written in specific personal styles; 3. Ability to answer multi-hop questions; 4. Retention of unmodified knowledge.
The authors used WikiBigEdit to test the performance of mainstream knowledge editing methods against general approaches in knowledge updating tasks. Experimental results demonstrate that models utilizing RAG methods achieved the highest accuracy on the test sets, while continued fine-tuning and model merging yielded second-best results, whereas knowledge editing methods performed unsatisfactorily. The findings indicate that existing knowledge editing techniques have significant limitations when applied to large-scale knowledge editing.

**Claims And Evidence:**

The authors designed comprehensive experiments to substantiate their arguments.

**Essential References Not Discussed:**

No missing key references were identified.

**Experimental Designs Or Analyses:**

In the experimental section, the authors evaluated the effectiveness of various knowledge-updating methods across different LLMs (such as LLaMA and others). For each method, the authors provided detailed analyses of their performance on different datasets and presented several significant conclusions.

**Methods And Evaluation Criteria:**

Regarding data, the authors employed a convincing approach to construct a large-scale dataset for testing whether models can retain the latest knowledge from Wikipedia. The test sets they developed comprehensively evaluate knowledge editing effectiveness from four distinct perspectives.
In terms of model selection, the authors tested several mainstream knowledge editing methods (including ROME, R-ROME, MEMIT, and WISE) as well as general-purpose RAG methods, continued fine-tuning approaches, and model merging techniques.

**Other Comments Or Suggestions:**

The main focus of this paper is to analyze the limitations of existing knowledge editing methods. However, in the experimental section, the authors merely replicated similar experiments from previous literature (only changing the dataset) and obtained similar results. It would be desirable for the authors to conduct a more in-depth analysis of the specific reasons why existing knowledge editing techniques underperform compared to general methods.

**Other Strengths And Weaknesses:**

Strengths:
- The authors constructed a large-scale and comprehensive knowledge editing dataset that effectively measures model performance on knowledge updating tasks.

**Questions For Authors:**

No questions.

**Relation To Broader Scientific Literature:**

This paper summarizes and emphasizes findings from existing literature, namely: as the number of knowledge entries to be edited increases, the ability of models trained with current knowledge editing techniques to memorize knowledge gradually deteriorates. Furthermore, through comparative experiments, this study reveals that knowledge editing techniques cannot outperform general methods (such as RAG, fine-tuning, etc.) and do not offer significant cost advantages.

**Theoretical Claims:**

No theoretical contribution was presented by the authors.

---

> ### Author Rebuttal · Authors · 2025-04-01
>
> We thank the reviewer for the thoughtful and constructive review. We are pleased that the reviewer found the motivation and contributions of our work to be of practical importance — particularly our effort to investigate the limitations of existing knowledge editing techniques under realistic conditions. We appreciate the recognition of our automated pipeline for extracting real-world knowledge updates, as well as the comprehensive evaluation framework of *WikiBigEdit*, which tests factual retention, generalization, and reasoning across multiple dimensions. We are also grateful for the acknowledgment of our systematic comparison of knowledge editing methods with general update approaches such as RAG and continual finetuning. We address the reviewer’s suggestion for deeper analysis and provide further clarifications below.
>
> **In-Depth Analysis of Knowledge Editing Failure**
>
> While our study builds on the foundations of previous work, our experiments go beyond a simple dataset substitution. Specifically, we design a setting that reflects real-world, large-scale lifelong knowledge editing, a departure from the typically synthetic and small-scale settings used in prior studies (see also related works). Our benchmark introduces several important new dimensions: (1) the sequential nature of edits over time, (2) the use of real-world factual updates derived from Wikidata, and (3) a comprehensive evaluation protocol that includes not just edit accuracy but also generalization, locality, and multi-hop reasoning.
>
> Moreover, our analysis compares knowledge editing methods not only to one another but also to broader model update strategies, such as retrieval-augmented generation (RAG) and continual finetuning with adapter merging. This allows us to contextualize the limitations of editing methods in a practical deployment landscape - a comparison absent in prior work.
>
> We agree that a deeper investigation into the specific mechanisms of failure in knowledge editing is an important direction. While our primary focus is on empirically establishing limitations at scale, we highlight recent work that examines these mechanisms in detail (see Section 2). For example, Hsueh et al. [1], Gupta et al. [2], and Yang et al. [3] demonstrate that even small-scale editing can induce instability or forgetting in LLMs. Gupta et al. [4] further attribute performance collapse during sequential edits to the lack of regularization, which leads to overfitting on individual updates. These insights complement our findings and underscore the challenges in designing robust editing techniques that scale.
>
> We will clarify this distinction and discussion in Section 2 of the revision.
>
> [1] Hsueh et al., Editing the Mind of Giants, 2024.
>
> [2] Gupta et al., Model Editing at Scale Leads to Gradual and Catastrophic Forgetting, 2024.
>
> [3] Yang et al., The Butterfly Effect of Model Editing, 2024.
>
> [4] Gupta et al., Lifelong Sequential Knowledge Editing without Model Degradation, 2025.

---

> > ### Comment · Reviewer_j8NE · 2025-04-07
> >
> > Thank you for your response, I will keep my rating unchanged.

---

### Official Review · Reviewer_Qubt · 2025-03-24

**Overall Recommendation:** 4

**Summary:**

This paper proposes a large-scale knowledge editing benchmark *WikiBigEdit* based on the Wikidata, which contains over 500k question-answer pairs. At the same time, this work also constructs a pipeline that can automatically update data to adapt to the changes in real data, while mitigating the overfitting problem of pre-training processes for large-scale language models on information before the training time point. After construction, this article tests the effectiveness of several mainstream methods on *WikiBigEdit*, reflecting various problems of previous methods in terms of locality, multi-hop, and personalized questioning (generalizability).

## update after rebuttal
I'll keep my assessment unchanged. The authors have answered my questions clearly.

**Claims And Evidence:**

Yes.
The text provides a detailed description of the construction process of the knowledge editing benchmark *WikiBigEdit*, and makes a detailed evaluation of the representative methods in the field on different open-source models, followed by analysis. The experimental results are very comprehensive.

**Essential References Not Discussed:**

There is no such situation in my vision. This is the first knowledge editing dataset to reach this scale and can be continuously updated. At the same time, as a benchmark, this work has compared several very mainstream knowledge editing methods.

**Experimental Designs Or Analyses:**

Yes, I have checked all the experiments, mainly including different models (7B level) under various knowledge editing methods on the *WikiBigEdit* benchmark proposed in this paper. Regarding the experiments, I have several questions I would like to ask the authors:
1. As described in the paper, in the Multi-Hop Reasoning test, the authors only considered connecting two triplets end-to-end (which is actually a two-step question). But is this too short for normal multi-hop question answering? General multi-hop question answering is usually based on a subgraph of a knowledge base or a longer reasoning chain.
2. Can some other results based on closed-source models  (considering GPT3.5, 4o-mini used in the experiment, the cost is actually not high) or larger open-source models (like Qwen-series) be supplemented in the experiment?

**Methods And Evaluation Criteria:**

The *WikiBigEdit* proposed in this paper is meaningful for knowledge editing tasks. On one hand, this is the first dataset of such a large scale constructed based on real data. On the other hand, the additional data update pipeline designed by the authors makes the benchmark keep pace with the times and avoids the overfitting or data leakage problems of newly emerged LLMs.

**Other Comments Or Suggestions:**

None

**Other Strengths And Weaknesses:**

As described in "Relation to Broader Scientific Literature" and "Experimental Designs or Analyses".

**Questions For Authors:**

1. The previous common sense knowledge graph (such as ConceptNet) is a relatively dirty knowledge base, where the same head entity and relationship can correspond to multiple tail entities. In the narrative of this paper, such relationships are directly removed. I would like to know what proportion is removed? Can there be some analysis experiments to analyze the quality of the extracted triples? (Because in my previous understanding, the dirty relationships or meaningless triples in such common sense graphs account for the majority.)
2. The construction of problems in the text is based on GPT3.5 or GPT-4o-mini. Have you tried other models for problem construction? Would the different distributions of different models lead to an impact on data quality?
Anyway, this is a good piece of work, and answering these questions will make my understanding more complete.

**Relation To Broader Scientific Literature:**

This paper proposes a large-scale dataset *WikiBigEdit* for the knowledge editing task. This dataset addresses the insufficient amount of data in previous knowledge editing tasks, as well as the issue of overfitting in the pre-training process after the emergence of large-scale language models. More importantly, this benchmark proposes a pipeline for continuously updating data, constantly updating the dataset based on reality, which very effectively solves the problem of models overfitting to all knowledge before a specific time point.

**Theoretical Claims:**

This paper is a work about a new benchmark and does not involve theoretical proof. All the formulas in the text merely describe the update process of model parameters and serve only as a narrative, not as proof.

---

> ### Author Rebuttal · Authors · 2025-04-01
>
> We thank the reviewer for their thoughtful and detailed evaluation. We appreciate the recognition of our key contributions: the construction of *WikiBigEdit* as the first large-scale, automatically extensible benchmark for real-world knowledge editing; the automated pipeline enabling continual updates; and the comprehensive evaluation across multiple LLMs and editing techniques. We are also grateful for the acknowledgment of our efforts to mitigate data leakage and pretraining overlap, as well as the clarity of our supplementary materials and code. We address the reviewer’s insightful questions below.
>
> **Length of Multi-hop Reasoning Chains**
>
> We appreciate the reviewer’s observation regarding the scope of multi-hop reasoning. While multi-hop can involve longer chains or subgraphs in general QA settings, in the context of knowledge editing, it is commonly defined as reasoning over two connected facts, forming a two-hop chain [1–3]. This is consistent with prior work, such as HotpotQA [4] and EX-FEVER [5], where two-hop questions are standard for evaluating a model’s ability to integrate and reason over multiple facts. Our benchmark adopts this convention to align with established protocols. That said, we agree that exploring longer reasoning chains is a valuable direction for future work.
>
> **Triplet Quality and Filtering**
>
> We thank the reviewer for their question on triple quality and filtering. To ensure a high-quality benchmark, our automated pipeline applies multiple filtering and QA steps (see Section 3.2). Below are average statistics from the first four timesteps:
> * Initial Input: 100% of new or changed factual triplets from consecutive Wikidata snapshots.
> * Initial Filtering: Remove cyclic, non-Roman, or overly long entities (~89% remain).
> * Filter Unwanted Relations: Keep only WikibaseItem relations (entity-to-entity), excluding media, URLs, etc., and a small list of low-value relations (e.g., “use”, “is a list of”), retaining ~39%.
> * Non-Deterministic Triplets: Remove (subject, relation) pairs with multiple objects (~24% remain).
> Final Filtering: Further filtered to remove spurious entries (final ~24% retained).
>
> We also conducted manual inspections, sampling triplets from various batches to confirm that relations and entity pairings are meaningful.
>
> These steps help ensure that only high-quality, interpretable factual triples are included in WikiBigEdit. We will extend Section 3.2 to report these statistics in the final version.
>
> **Impact of LLM Choice on QA Generation**
>
> We thank the reviewer for raising the important question of how the choice of LLM affects QA pair generation. In preliminary experiments, we evaluated the sensitivity of our generation pipeline to different GPT models.
>
> For the Update and Locality sets, model choice had a negligible impact. These questions are directly derived from subject–relation–object triplets, and even GPT-3.5-turbo consistently produced accurate and well-formed outputs. For example, given the triplet (SS Heliopolis, manufacturer, Fairfield Shipbuilding and Engineering Company), GPT-3.5-turbo generated “Which company was the main manufacturer of SS Heliopolis?”, GPT-4o-mini produced “Who was the manufacturer of the SS Heliopolis?”, and GPT-4o generated “Which company was the manufacturer of the SS Heliopolis?” — all of which are stylistically comparable and semantically correct.
> The Rephrase set showed similarly consistent quality across models, as paraphrasing proved to be a relatively simple task for most LLMs.
>
> In contrast, the Multi-hop and Persona sets were more sensitive to model choice, as these tasks require compositional reasoning or stylistic transformation. For instance, in the multi-hop case combining (2004 VS75, discoverer, Marc Buie) and (Marc Buie, gender, male), GPT-3.5-turbo generated the incomplete question “Who is the discoverer or inventor of (525257) 2004 VS75?”, failing to incorporate both facts. Meanwhile, GPT-4o-mini and GPT-4o correctly generated queries such as “What is the gender of the discoverer of 2004 VS75?”
> Based on these findings, we selected GPT-4o-mini for the Multi-hop and Persona sets, as it offers a strong balance between generation quality and efficiency. We observed no significant difference between GPT-4o and GPT-4o-mini in these settings.
> We hope this clarifies the robustness of our generation pipeline and our rationale for model selection. We will include an ablation on generation models in the Supplementary Material.
>
> [1] Cohen et al.,  Evaluating the Ripple Effects of Knowledge Editing in Language Models, 2023.
>
> [2] Zhong et al., MQuAKE: Assessing Knowledge Editing in Language Models via Multi-Hop Questions, 2023.
>
> [3] Zhong et al., MQuAKE-Remastered: Multi-Hop Knowledge Editing Can Only Be Advanced with Reliable Evaluations, 2025.
>
> [4] Yang et al., HotpotQA: A Dataset for Diverse, Explainable Multi-hop Question Answering, 2018.
>
> [5] Ma et al., EX-FEVER: A Dataset for Multihop Explainable Fact Verification, 2024.

---

### Decision · Program_Chairs · 2025-05-01

**Decision:**

Accept (poster)

**Comment:**

This paper introduces WikiBigEdit, a large-scale, continuously updating benchmark designed to evaluate the capabilities of knowledge editing in large language models (LLMs). Built from Wikidata, it comprises over 500,000 question-answer pairs generated through an automated extraction and rewriting pipeline. The benchmark spans multiple knowledge editing dimensions, including multi-hop reasoning, stylistic variation, and knowledge retention, enabling comprehensive evaluation of modern LLMs. Reviewers (Qubt, j8NE, UH8c) universally acknowledge the novelty and practical relevance of this benchmark, especially in light of real-world challenges such as outdated knowledge and the limitations of retraining. They also commend the meticulous data construction pipeline, strong empirical evaluation across a range of methods (e.g., ROME, MEMIT, RAG, fine-tuning), and clear experimental insights that reveal current weaknesses in existing knowledge editing techniques (e.g., poor scalability and generalization). The benchmark’s alignment with real-world data and its reproducibility—via shared code and data—further enhance its value to the community.

While the overall contribution is strong, reviewers suggest several areas for refinement. First, the benchmark’s multi-hop evaluation may be too simplistic, as it only considers two-hop reasoning; more complex chains could better represent real-world multi-step inference (Qubt). Additionally, there is limited exploration of whether different LLMs used in the QA construction phase (e.g., GPT-3.5 vs. GPT-4o-mini) introduce distributional biases that affect benchmark quality (Qubt). The current evaluation focuses largely on well-known open-source models, and broader comparison with larger or closed-source systems (e.g., Qwen or GPT-4) could offer deeper insights (Qubt, UH8c). Reviewers also recommend testing the benchmark against recent baselines (e.g., HippoRAG, DiRe tests) and concurrent datasets like TiC-LM for multi-hop and long-term reasoning comparisons (UH8c). Lastly, while j8NE appreciates the benchmark’s value, they point out that much of the experimental analysis replicates existing setups on a new dataset and suggest that more in-depth failure analysis would improve the interpretability of performance gaps. Despite these concerns, all reviewers agree that the benchmark is timely, impactful, and fills a critical gap in the evaluation of dynamic knowledge in LLMs.